# Collapse and resurgence of the Iceland mantle plume

**Callum Pearman** [1] ✉, **Chia-Yu Tien** [1], **Nicky White** [1], **John Maclennan** [2], **Bramley Murton** [3], **Sally Gibson** [2], **Jason Day** [2], **Ross Parnell-Turner** [4] & **IODP Expedition 395 Scientists***

Mantle plumes are a fundamental component of the Earth's convective regime, but their long-term behaviour remains poorly understood. The Iceland plume, which is bisected by the Mid-Atlantic Ridge, presents an opportunity to constrain plume evolution. Here, we reconstruct its influence upon seafloor spreading by exploiting the geochemistry of basalts drilled south of Iceland during International Ocean Discovery Program (IODP) Expedition 395. Trace element and isotopic measurements, combined with contextual geophysical observations, demonstrate that plume influence waned after continental break-up at ~ 55 Ma, collapsed rapidly at ~ 38 Ma, and then was progressively re-established to the present day. Recovered ~ 32 Ma basalt samples have rare earth element compositions equivalent to mid-Atlantic Ridge dredge samples located south of the present-day plume influence. These compositions can be modelled by passive upwelling and melting of depleted MORB mantle with a potential temperature of ~ 1300 °C. In contrast, basalts recovered from younger (i.e. 0–14 Ma) sites show unequivocal evidence for plume influence. Together, these results imply dramatic changes in the extent of plume-ridge interaction across the North Atlantic region, providing key chemical constraints for geodynamic models of plume evolution and its imprint upon the geological record.

Mantle plumes are a key component of mantle convection within the Earth[1,2]. Plumes cause surface uplift, generate ocean island volcanism, enhance melting at mid-ocean ridges, and sometimes control oceanographic gateway elevation[3,4]. A recent study suggests that there may be up to 18 globally distributed and long-lived mantle plumes, although their temporal evolution remains poorly constrained[2].

The North Atlantic Ocean presents an important opportunity to investigate plume evolution since outflow of the Iceland plume is directly sampled by mid-ocean ridge melting along the Reykjanes Ridge south of Iceland. To exploit this opportunity, International Ocean Discovery Program (IODP) Expedition 395 drilled into 100–200 m of oceanic basement at five different sites located along a flowline transect (i.e. oriented parallel to plate spreading direction) east of the Reykjanes Ridge at a radial distance of ~740 km from the present-day centre of the plume (Fig. 1). Four drilling sites targeted V-shaped ridges and troughs (VSRs and VSTs), which represent crustal thickness variations that relate directly to plume–ridge interaction. VSRs are suspected to form by sampling of thermal mantle anomalies advecting away from a pulsing Iceland plume[5–7]. The drilling sites are positioned along a regional seismic reflection profile, JC50-1, that has been described and interpreted in detail elsewhere[8]. Throughout the North Atlantic Ocean, the age of the ocean floor is constrained by

[1]Bullard Laboratories, Department of Earth Sciences, University of Cambridge, Cambridge, UK. [2]Department of Earth Sciences, University of Cambridge, Cambridge, UK. [3]National Oceanography Centre, Southampton, UK. [4]Institute of Geophysics & Planetary Physics, Scripps Institution of Oceanography, University of California, San Diego, CA, USA. *A full list of members and their affiliations appears in the Supplementary Information.
✉ e-mail: cp782@cam.ac.uk

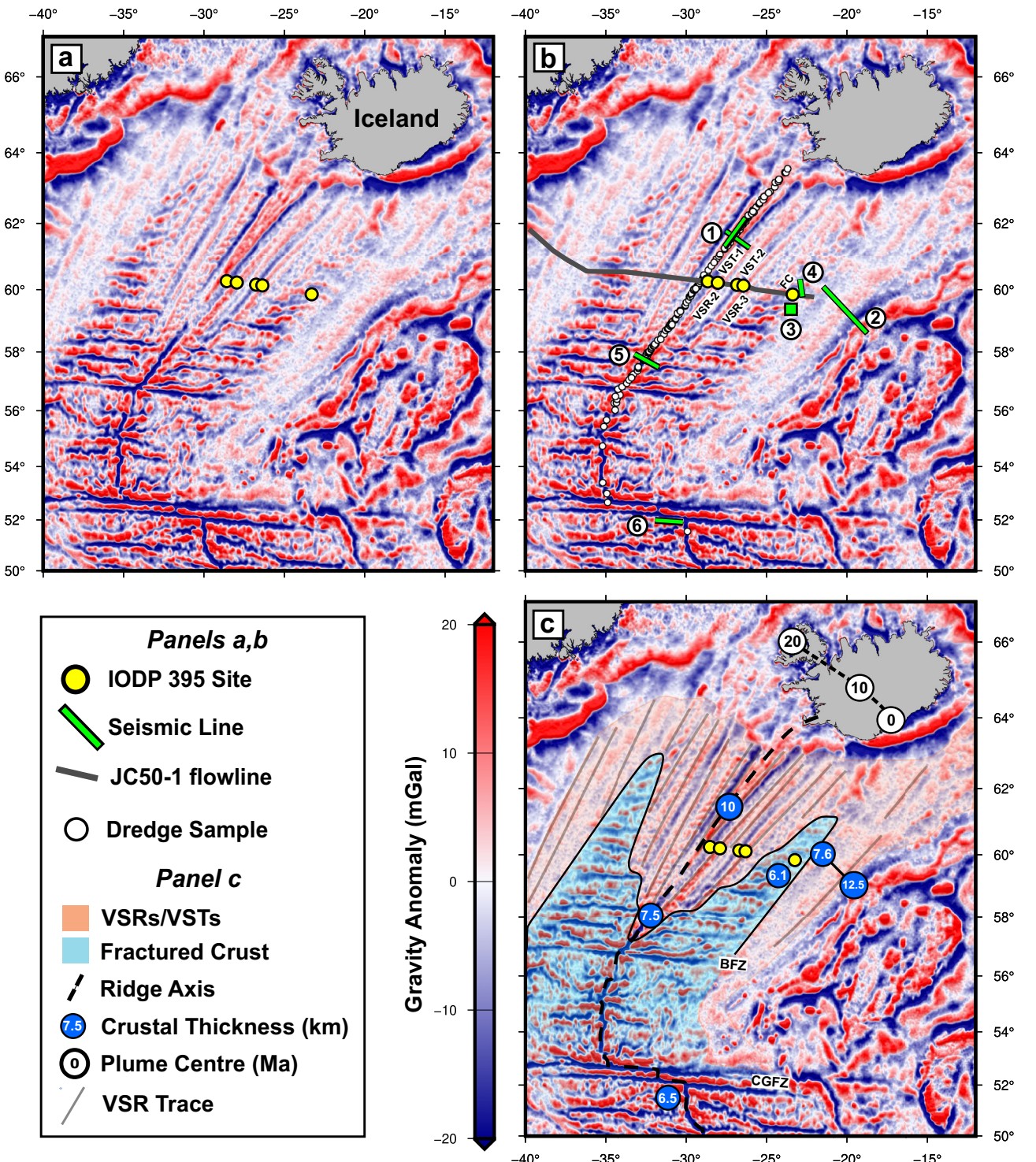

**Fig. 1 | Geological setting of IODP Expedition 395. a** High-pass free-air satellite gravity map of North Atlantic Ocean filtered to remove wavelengths >100 km[69]. Yellow circles = IODP Expedition 395 sites (note scale bar beside (**c**)). **b** Same showing location of legacy/modern seismic refraction/wide-angle surveys and locations of published dredge samples. Numbered green lines = seismic surveys (1[20]; 2[42]; 3[21]; 4[22]; 5[27]; 6[23]); white circles = dredged basalt and glass samples[15–17]; dark grey flowline that is parallel to plate spreading direction[9] = JC50-1 seismic reflection profile[43]; labelled yellow circles = IODP sites as before. **c** Same with the interpretation of oceanic crustal type and thickness. Orange/blue colours = smooth/fractured crust; numbered blue circles = crustal thickness values from surveys shown in (**b**); yellow circles = IODP sites; numbered white circles with connecting dashed black line on Iceland = reconstructed locus of plume conduit as function of time in Myrs[51]; thicker black dashed line = locus of Reykjanes Ridge and its continuation toward mid-Atlantic Ridge; grey lines = traces of significant V-shaped ridges; BFZ Bight Fracture Zone, CGFZ Charlie Gibbs Fracture Zone.

extensive aeromagnetic and ship-board magnetometer surveys[9,10]. Additional calibration is provided by downhole magnetostratigraphic measurements together with recovered palaeontological assemblages from the base of each sedimentary succession[11].

Analyses from boreholes at the five relevant sites are presented in this study. The four boreholes that penetrate V-shaped ridges and troughs are: U1555I (i.e. VST-1) with a magnetic basement age of 2.8 Ma; U1563B (VSR-2) at 5.2 Ma; U1554F (VST-2) at 12.7 Ma; and U1562B (VSR-

3) at 13.9 Ma (Fig. 1a, b). A fifth borehole (U1564F hereinafter referred to as Site FC, where FC stands for fractured crust) targeted much older (i.e. 32.4 Ma) rugose crust that is devoid of V-shaped ridges and troughs but instead is characterised by extensive fracture zones (Fig. 1a, b). The development of these approximately symmetrical lobes that are dominated by fracture zones is subject to debate. For example, it has been suggested that transform fault formation at the mid-oceanic ridge was triggered by a switch from orthogonal to oblique seafloor spreading[12]. Alternatively, these fractured lobes may indicate that the planform of the previously large and hot Iceland plume either dramatically shrank or withdrew[6]. Reversion to smooth (i.e. not fractured) crustal generation with development of V-shaped ridges and troughs may imply plume resurgence with concomitant increase in volume flux from ~30 Ma to the present day[4], migration of the Reykjanes Ridge towards the plume conduit[13] or unrelated structural reorganisation[10].

Here, we wish to test these alternative hypotheses by exploiting trace element and radiogenic isotopic measurements from all five sites. These measurements are combined with crustal thickness constraints from modern seismic wide-angle and legacy refraction surveys in order to reconstruct the influence of the Iceland plume during Cenozoic times. This multi-disciplinary approach should enable us to test the predictions of different geodynamical models of long-term plume behaviour and to evaluate the extent to which this behaviour has affected North Atlantic crustal architecture.

We propose that the Iceland plume influence first started to wane after continental break-up at ~55 Ma and then rapidly collapsed, receding to within 500 km of the putative plume centre by 32 Ma. This proposal is consistent with the classical head-tail geometric development of a deep-seated mantle plume[14]. At ~30 Ma, plume influence started to re-establish itself, which may reflect migration of the mid-ocean ridge towards the plume conduit[4].

## Results

### Geochemical composition of 32 Ma fractured crust

Rare earth element (REE) concentrations of basalt glass samples from all five sites were analysed by LA-ICP-MS (see 'Methods'). Whole-rock samples from the FC site were analysed by ICP-MS. Major element concentrations were analysed by EPMA and XRF. Our REE measurements were combined with published REE measurements of axial dredge samples located between 51.5°N and 64°N that were compiled from the legacy datasets of Gale et al.[15], Murton et al.[16] and Jones et al.[17] (see Source Data).

Compositions of axial dredge samples reveal an approximately linear trace element enrichment gradient that extends from the Reykjanes Peninsula to ~61.2°N (Fig. 2a). South of 61.2°N, basalt samples are depleted in incompatible trace elements relative to globally averaged N-MORB and D-MORB values[15,18]. This depletion pattern persists as far south as the mid-Atlantic Ridge 60 km south of Charlie Gibbs Fracture Zone (CGFZ) at ~51.6°N[15]. FC glass samples have REE ratios (e.g. Ce/Yb) that are enriched relative to the Reykjanes Ridge south of 61.2°N but similar to mid-Atlantic Ridge samples at ~51.6°N and samples within the Icelandic enrichment gradient at ~62°N (Fig. 2a).

Principal component analysis (PCA) of the combined dataset indicates that two eigenvector components, $P_1$ and $P_2$, can account for 98% of observational variance. These two components reflect changes in the concentration and in the slope of the REE distributions, respectively. They have associated eigenvalues of $\lambda_1 = 77\%$ and $\lambda_2 = 21\%$. Significantly, distinct compositional clusters occur in rotated principal component space (Fig. 2b). Trace element-depleted basalts from the Reykjanes Ridge form a linear array. However, samples located north of ~61.2°N diverge with respect to La-Sm vectors towards higher $P_{2r}$ scores that reflect mantle source enrichment, where the subscript $r$ indicates rotated space. Notably, the average glass composition together with the bulk of whole-rock samples at Site FC, occupy the

same distinct PC space as both samples from the mid-Atlantic Ridge at 51.6°N and global D-MORB. This similarity of REE composition is quantifiable by measuring the distance in $P_{1r}$-$P_{2r}$ space between axial dredge samples located between 51.5°N and 64°N and the glass composition at Site FC. This distance approaches zero at ~51.6°N (Fig. 2d). Note that samples from Site FC and from the mid-Atlantic Ridge at 51.6°N are offset along light REE vectors from the linear Reykjanes Ridge array, approximately coincident with the locus of global D-MORB composition. This offset implies that the mantle source supplying the Reykjanes Ridge is relatively depleted[19].

Co-location within $P_{1r}$-$P_{2r}$ space of Site FC samples and mid-Atlantic Ridge samples at 51.6°N reflects uniform REE enrichment compared with samples which lie within the Icelandic enrichment gradient at ~62°N. This uniform enrichment, represented by an increase in $P_{1r}$ score, can be modelled by a decrease in mantle potential temperature of 50±25 °C or, alternatively, by 19% of olivine crystallisation (Fig. 2c). However, samples at ~62°N, 51.6°N together with Site FC samples have similar average MgO contents (i.e. 7.7–7.8 wt%), which precludes significantly different extents of fractional crystallisation. Therefore, differences in REE concentrations most likely reflect lower melt fractions at Site FC and at the mid-Atlantic Ridge near 51.6°N. Lower cumulative melt fractions are corroborated by the results of modern seismic wide-angle and legacy refraction surveys, which show that oceanic crustal thicknesses near Site FC and just south of 52°N are less than the crustal thickness of 10 ± 1 km recorded at 62°N[20] (Fig. 1c). At Site FC, two legacy seismic refraction surveys show that oceanic crust is no more than 6.1 ± 1.0 km[21,22]. At the mid-Atlantic Ridge near 52°N, a modern wide-angle survey shows that oceanic crust is 6.5 ± 0.7 km[23]. These values lie within the global mean crustal thickness of 6.38 ± 1.12 km and are consistent with ambient asthenospheric temperatures[24].

The position of the Charlie Gibbs Fracture Zone approximately marks the southernmost extent of anomalously elevated bathymetry associated with the Iceland plume swell (Fig. 2d). Thus, samples from the mid-Atlantic Ridge at 51. 6°N, where oceanic crust thickness is close to the global average value, appear to represent ambient, plume-free melting conditions. This location is a sufficient distance of ~60 km from the active transform fault such that mid-oceanic ridge melting is unlikely to be influenced by local lithospheric cooling[25]. The similarity of REE concentration profiles at Site FC and at 51.6°N suggests that the mantle melting at Site FC was equally unaffected by the Iceland plume, despite its proximity to the inferred position of the plume conduit at that time. This similarity will be tested by quantitative modelling in Section 'Geochemical modelling at site of fractured crust'.

### Radiogenic isotopic constraints for mantle source composition

Along the Reykjanes Ridge, an observed gradient of radiogenic isotopic enrichment is usually interpreted as progressive dilution of an enriched plume source by mixing with depleted mantle during southward outflow from Iceland[26] (Fig. 3a). However, this mechanism alone cannot account for the extreme incompatible trace element depletion observed south of ~61.2°N. Equally, elevated degrees of in situ mantle melting fail to account for this pattern since trace element depletion persists even where oceanic crustal thickness values approach ~7.5 km just north of the Bight Fracture Zone (Fig. 1b, c)[27]. Instead, this axial trend is more plausibly accounted for by a depleted mantle component that is intrinsic to the Iceland plume itself, combined with exhaustion of enriched mantle heterogeneities during progressive melting along the ridge axis[16,28,29].

To gauge whether or not this pattern of depletion has remained stable as a function of time, radiogenic Nd isotopic compositions of whole-rock samples from each of the IODP sites have been measured and compared with an axial dredge dataset compiled from Blichert-Toft et al.[30], Jones et al.[17] and Thirlwall et al.[31]. If the mantle source remained unchanged, our measurements would exhibit $\varepsilon_{Nd}$ values

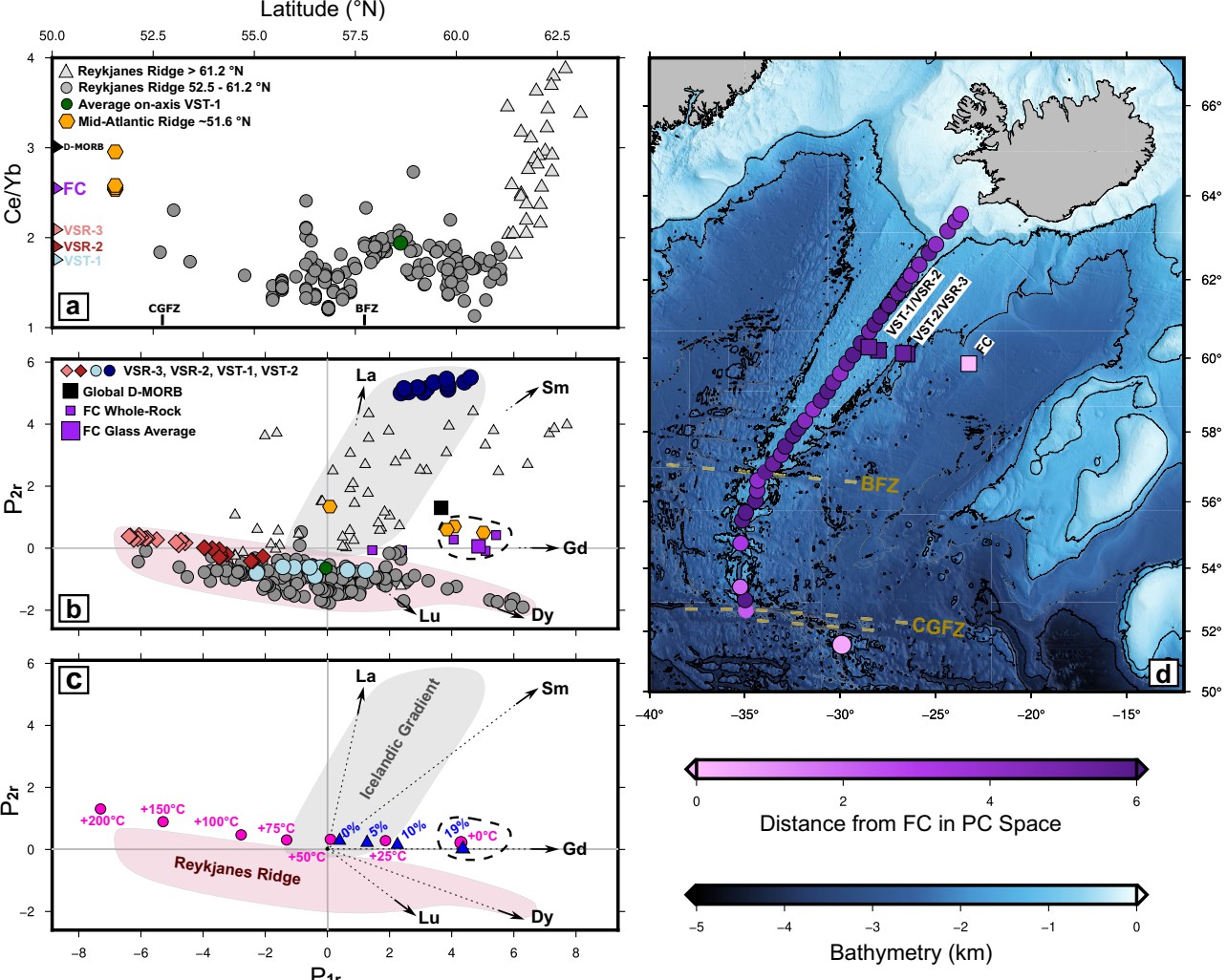

**Fig. 2 | Comparison of site FC rare-earth element composition to Reykjanes Ridge. a** Ce/Yb ratios of samples plotted as a function of latitude along Reykjanes Ridge. Average Ce/Yb of Site VST-2 is 5.8. Light/dark grey triangles/circles = legacy analyses along the Reykjanes Ridge north and south of 61. 2°N, respectively[15–17]; green circle = locus of average ratio for on-axis VST-1; orange hexagons = loci of ratios from the mid-Atlantic Ridge at 51. 6°N; coloured labels (D-MORB, FC, VSR-3, VSR-2 and VST-1) = average ratios for depleted mid-ocean ridge basalts[15], Site FC glass, and for V-shaped ridge/trough sites. **b** Loci of analyses as a function of rotated principal components $P_{1r}$ and $P_{2r}$ where scores and loadings are rotated to ensure that uniform REE enrichment/depletion plots along the horizontal axis. Pink/red diamonds and turquoise/blue circles = downhole analyses of glasses from VSR-3/VSR-2 and from VST-1/VST-2; small purple and large purple squares = whole-rock and average glass analyses for Site FC; black square = global value for D-MORB[15]; labelled black arrows = vectorial orientations for La, Sm, Gd, Dy and Lu;

grey/pink polygons = dominant loci of analyses that define Icelandic/Reykjanes Ridge gradients; black dashed line = locus of inferred non-plume analyses; other symbols are the same as in (**a**). **c** Same as (**b**), showing results of pyMelt forward modelling. Labelled pink circles = loci of modelled melt compositions for temperature anomalies up to + 200 °C; labelled blue triangles = loci of calculated melt models for different amounts of olivine crystallisation. **d** Bathymetric map of the North Atlantic Ocean, which summarises distance in principal component space. Bathymetry data are the GEBCO Compilation Group (2025) GEBCO 2025 Grid (doi:10.5285/37c52e96-24ea-67ce-e063-7086abc05f29)[70]. Pink-to-purple circles/squares = loci of analyses where colour is indicative of distance in $P_{1r}$-$P_{2r}$ space away from the average composition of Site FC glass, where zero means identical; black line = 2 km bathymetric contour; gold dashed lines = major fracture zones as before.

comparable to those of axial samples that are intersected by the flowline along which the drilling sites are located (i.e. $\varepsilon_{Nd}$ = 10.1 ± 0.5)[26,30,31] (Fig. 3a). Instead, the average value of $\varepsilon_{Nd}$ increases from 7.4 to 10.1 as a function of time (Fig. 3b).

At sites located on the older ridge-trough pair (i.e. VST-2 and VSR-3), $\varepsilon_{Nd}$ values are low, which indicates the presence of a greater proportion of a mantle source component with low $^{143}Nd/^{144}Nd$ ratios, such as pyroxenite[32–34]. Assuming the trace element and Nd isotopic compositions of a typical lherzolite and of KG-1 pyroxenite described in Supplementary Fig. 1, the observed increase in $\varepsilon_{Nd}$ towards the ridge axis can be matched by invoking a ~10–17 % decrease in the contribution of pyroxenitic melt to oceanic crustal formation as a function of time towards the present day (Fig. 3b). This range of values

corresponds to a ~2–5 % decrease in the contribution of pyroxenite to the mantle source.

At Site FC, $\varepsilon_{Nd}$ = 9.5 ± 0.4 (Fig. 3b). This value falls within the range of typical Atlantic MORB estimates ($\varepsilon_{Nd}$ ~ 9–13[28,35]). It is therefore consistent with the absence of a mantle plume component. However, a value of 9.5 alone cannot unambiguously exclude plume influence since it is comparable to the value obtained for samples at ~62°N along the Reykjanes Ridge (i.e. proximal to Iceland). These dredge samples also differ in respect to other proposed source proxies such as ΔNb, which has a value of ~0.2 at 62°N and a value of 0.00 ± 0.03 at Site FC[36]. There are also fundamental differences in REE concentrations and in crustal thickness (Figs. 2b and 1b). A non-plume depleted mantle component from the North Atlantic region might be expected to have

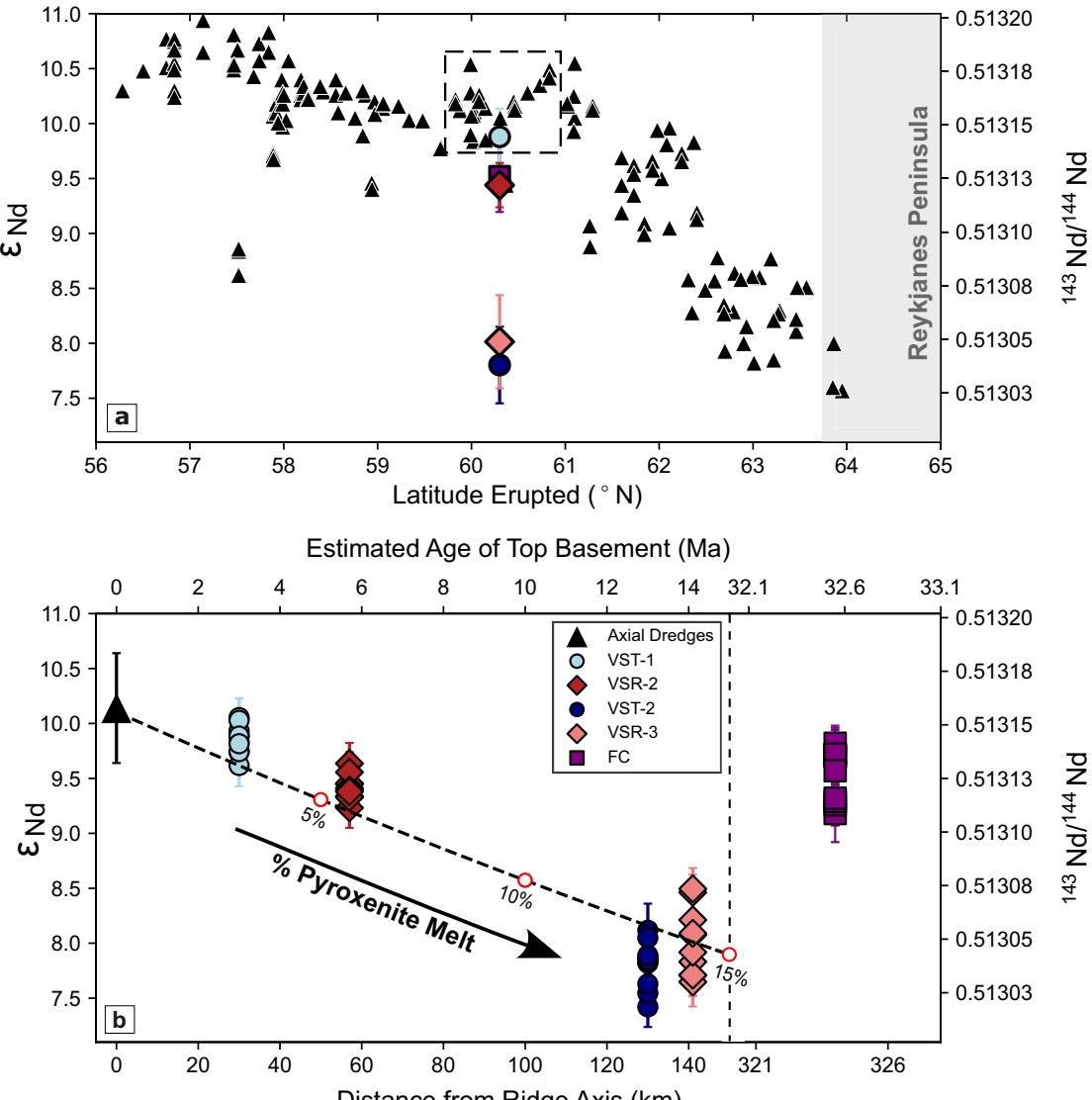

**Fig. 3 | $^{143}$Nd/$^{144}$Nd measurements of samples. a** $^{143}$Nd/$^{144}$Nd and $\varepsilon_{Nd}$ plotted as a function of latitude. Black triangles = compilation of legacy analyses[17,30,31] (black triangles at 57.5°N = analyses for enriched seamount 14D[16]); pink/red diamonds and turquoise/blue circles with vertical bars = averaged and projected measurements for sites VSR-3/VSR-2 and VST-1/VST-2 (see 'Methods' for further details); vertical bars represent borehole range; dashed box = locus of axial measurements used to estimate average value of $\varepsilon_{Nd}$ = 10.14 at the intersection of flowline and mid-oceanic ridge; vertical grey bar = onshore portion of Reykjanes Ridge on Reykjanes Peninsula. **b** $^{143}$Nd/$^{144}$Nd and $\varepsilon_{Nd}$ plotted as a function of distance from the ridge axis along

the flowline and as a function of age, assuming a half-spreading rate of 1 cm/yr. Black triangle with vertical bar = average value for axial dredge samples located within the dashed box shown in (**a**), where the vertical bar represents the data range; turquoise/red/blue/pink circles/triangles/squares with vertical bars = values for VST-1/VSR-2/VST-2/VSR-3/FC sites, at each of which 10 downhole measurements were made (note that 2 values for Site FC, with very large uncertainties, are excluded here but included in Source Data); dashed line with labelled open red circles = calculated variation of $\varepsilon_{Nd}$ as a function of distance/age, assuming a percentage increase of added pyroxenite melt component.

a ΔNb value of ~0, which would represent an intermediate composition between enriched Icelandic and depleted Reykjanes Ridge endmembers. Therefore, we conclude that the radiogenic isotope composition at Site FC, in combination with other geochemical and geophysical observations, is consistent with a plume-free mantle source.

At ~51.6°N along the mid-Atlantic Ridge, $\varepsilon_{Nd}$ values are typically >10.5, which suggests that the mantle source at Site FC was slightly more enriched[15]. It is important to emphasise that these differences in $\varepsilon_{Nd}$ are small and fall within the expected range of small-scale heterogeneity of plume-free mantle. Even though these widely used source proxies vary significantly along the Reykjanes Ridge, it is generally accepted that the thermal structure of the Iceland plume has influenced both crustal thickness and composition on a planform which has a present-day radius of ~1000 km[16,17,20,36]. Thus, source

proxies may not necessarily be reliable indicators of the presence or absence of plume influence. Instead, we suggest that the depth and degree of isentropic melting at the mid-oceanic ridge, which can be inferred by combined modelling of trace element concentrations and oceanic crustal thickness, is a clearer indicator of plume influence.

### Geochemical modelling at site of fractured crust
To determine the depth and degree of melting at Site FC, we exploit a forward modelling approach that uses isentropic melting paths to calculate REE concentrations, which are then compared with observed REE concentrations. The pyMelt Python library[37] builds upon the original polyphase fractional melting model of McKenzie and O'Nions[38]. Recovered melt fraction as a function of depth is used to infer mantle potential temperature by assuming the empirical parameterisation of

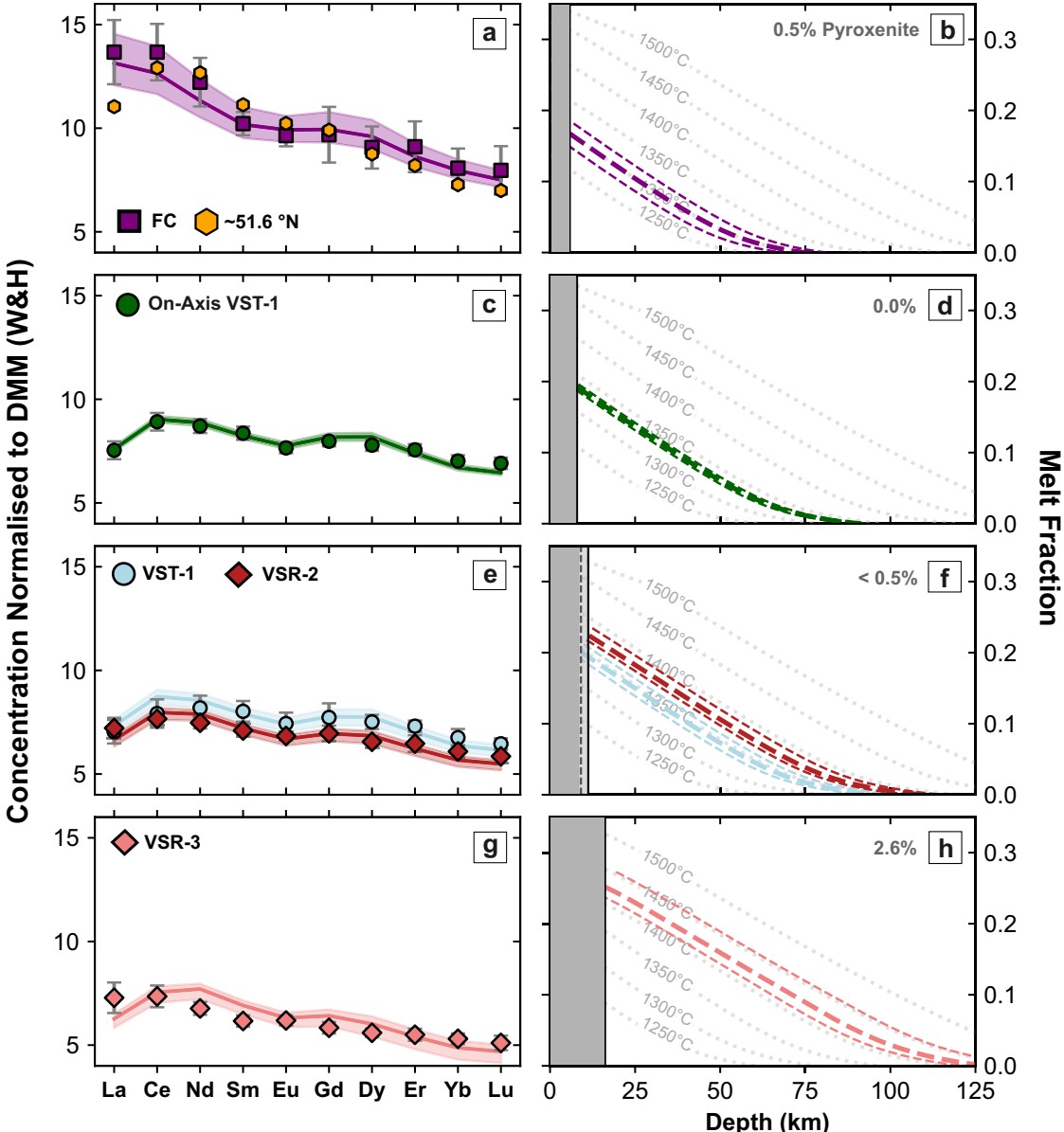

**Fig. 4 | Forward modelling of REE concentrations. a** Observed and calculated REE concentrations normalised with respect to average depleted MORB mantle (DMM)[40]. Purple squares with vertical bars = average values ±1$\sigma$ of REE concentrations obtained from glass samples at Site FC and corrected for fractional crystallisation; orange hexagons = same for glass samples from the mid-Atlantic Ridge at 51.6°N; purple line with band = REE concentrations calculated using the pyMelt package for a range of melt fractions as a function of depth shown in (**b**), assuming a mantle source enriched by 0.5% pyroxenite[37]. **b** Melt fraction plotted as a function of depth. Purple dashed and dotted lines = range of adiabatic melt fraction

Katz[39]. Here, we assume that there are two mantle lithologies which undergo melting—lherzolite containing trace amounts of water[39] and a KG-1 silica-deficient pyroxenite[37] whose relative proportions are determined using the measured value of $\varepsilon_{Nd}$.

Figure 4 a, b demonstrates that the average glass composition at Site FC can be satisfactorily modelled by assuming passive upwelling and melting of lherzolite that has a depleted MORB mantle composition (i.e. average value of DMM estimated by Workman and Hart[40]). Since $\varepsilon_{Nd}$ = 9.5 ± 0.4, a small component (i.e. 0.5%) of KG-1 pyroxenite has been included. The recovered distribution of melt fraction as a function of depth is consistent with a mantle potential temperature of 1298 ± 15 °C which yields a crustal thickness (i.e. cumulative melt

distributions used to calculate REE concentrations in (**a**); labelled grey dotted lines = melt fraction as a function of depth obtained by melting of hydrous lherzolite over the indicated range of potential temperatures using the parameterisation of Katz et al.[39]; grey rectangle indicates oceanic crust, whose base represents the top of the melting column. **c, d** Same for whole-rock samples from on-axis VST-1, assuming no pyroxenite component[16,17]. **e, f** Same for glass samples from VST-1 and VSR-2, assuming <0.5% pyroxenite component. **g, h** Same for glass samples from VSR-3, assuming 2.6% pyroxenite component.

thickness) of 5.9 ± 1.0 km. This result is consistent with a measured crustal thickness of 6.1 ± 1.0 km obtained from a nearby legacy seismic refraction survey[21] (Fig. 1b, c). At 51. 6°N on the mid-Atlantic Ridge, REE concentrations are similar to those measured at Site FC, albeit with slight depletion of heavy REEs and a negative La anomaly (Fig. 4a). Forward modelling of dredged glasses at 51.6°N yields a mantle potential temperature of 1305 ± 10 °C and a crustal thickness of 6.2 ± 0.6 km. This estimate of crustal thickness agrees with the value of 6.5 ± 0.7 km obtained by waveform modelling of a nearby modern seismic wide-angle survey[23] (Fig. 1b, c).

From a global perspective, the ambient asthenospheric potential temperature beneath the mid-oceanic ridge system is estimated from

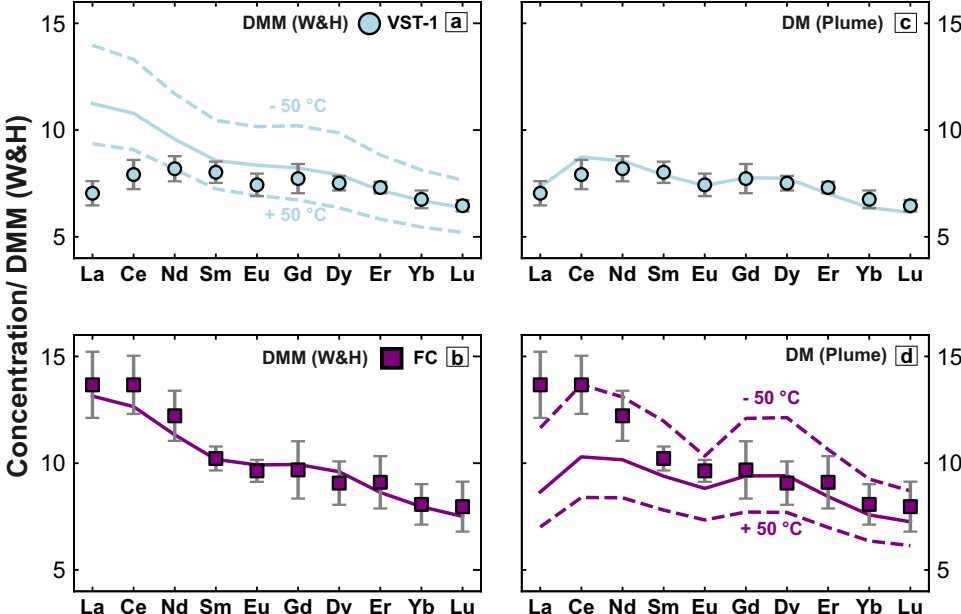

**Fig. 5 | Testing mantle source compositions. a** Observed and calculated REE concentrations normalised with respect to average depleted MORB mantle (DMM)[40]. Turquoise circles with vertical bars = average values ±1σ of REE concentrations for VST-1 corrected for fractional crystallisation; solid and dashed lines = REE concentrations calculated by pyMelt forward modelling, assuming an average DMM source of Workman and Hart[40] for different melt fractions as a function of depth, corresponding to a potential temperature range of 1342 ± 50 °C. **b** Same for Site FC with a temperature range of 1298 ± 50 °C. **c, d** Forward modelling of VST-1 and Site FC assuming a depleted mantle source obtained by calibration for plume-influenced North Atlantic mantle (see main text for further details).

petrological and geochemical studies to be 1332 ± 18 °C[24]. Our results for both Site FC and the mid-Atlantic Ridge at 51.6°N fall within this range and also agree with the global average value of oceanic crustal thickness. Notwithstanding minor differences in the degree of enrichment of their respective mantle sources, it is reasonable to conclude that melting at both sites occurred by passive upwelling of mantle material in the absence of a plume-driven thermal anomaly. This conclusion confirms that the Iceland mantle plume head was considerably smaller by ~32 Ma and had a limited influence upon the North Atlantic mantle source region compared with the present day. This result contradicts suggestions that the size and influence of the Iceland plume remained stable throughout the period of North Atlantic seafloor spreading[10,12].

### Plume influence on V-shaped ridges and troughs

The REE concentrations of VST-1, VSR-2 and VSR-3 are similar to those of samples along the Reykjanes Ridge south of ~ 61.2°N (Fig. 2a). In contrast, basalt glasses from borehole U1554F (VST-2) have enriched trace element signatures that closely match those of the Reykjanes Peninsula. Therefore, borehole U1554F may have penetrated an isolated seamount or been influenced by localised mantle heterogeneity, which means that its composition might not be representative of this particular V-shaped trough[19]. It is not considered any further in this contribution.

PCA indicates that sites VST-1, VSR-2 and VSR-3 lie within the Reykjanes Ridge linear array (Fig. 2b, c). This array is primarily aligned with the Gd-Dy-Lu vectors, which suggests that translation along the array reflects a combination of changes in depth of melting that affects heavy REEs and degree of melting and/or fractional crystallisation (Fig. 2c).

REE concentrations at these three sites cannot easily be modelled by assuming the depleted MORB mantle source (DMM) estimated by Workman and Hart[40], which was successfully employed at Site FC (Fig. 5). It is clear that the DMM source is too enriched in light REEs (e.g. La, Ce and Nd), and so an appropriately more depleted mantle source, which provides a better match between observed and calculated REE

concentrations together with observed crustal thickness for the on-axis V-shaped trough, was determined by calibration (Fig. 4c, d)[19]. This calibrated source composition is probably a closer approximation of the intrinsic depleted component of the Iceland plume[28]. Here, we use it to constrain the lherzolite component at each site, whilst the proportion of the KG-1 pyroxenite component is determined from the observed value of $\varepsilon_{Nd}$.

The results of forward modelling these three V-shaped ridge and trough sites are presented in Fig. 4e–h. In each case, a good match between observed and calculated REE concentrations is obtained by assuming that melt fraction varies adiabatically with depth for temperatures which exceed that of the ambient asthenosphere. VST-1 and VSR-2 require temperatures of $1342^{+13}_{-10}$ °C and $1373^{+14}_{-8}$ °C, respectively. Such values are consistent with the location of these sites towards the leading edge of the Iceland plume head. Recovered crustal thickness values of $8.9^{+1.0}_{-0.8}$ km and $11.5^{+0.9}_{-0.7}$ km are consistent with both observed thicknesses and with residual depth measurements[20]. It is reasonable to conclude that V-shaped ridges and troughs are generated by minor (i.e. ±25 °C) thermal fluctuations within the radially expanding plume head. VSR-3 has a higher temperature of $1420^{+28}_{-15}$ °C.

The difference between the DMM source component used for Site FC and the more depleted source component required for the VSR/VST sites suggests that plume influence can increase without being determinable by $\varepsilon_{Nd}$. Furthermore, it is clear that the melt fraction as a function of depth differs significantly between Site FC and the flanks of the Reykjanes Ridges, where V-shaped ridges and troughs are observed. This observation is best accounted for by a long-period oscillation of mantle potential temperature that is a manifestation of plume collapse and resurgence across the North Atlantic region.

## Discussion

Following continental break-up at ~56 Ma[9,41], plume-related magmatism became widespread and generated excessively thick oceanic crust (e.g. ~12.5 km at ~52 Ma)[42] (Fig. 6a). By ~40 Ma, oceanic crustal thickness decreased to 7.6 ± 0.2 km, which reflects a waning of plume influence. This thickened crust is smooth and appears to exhibit

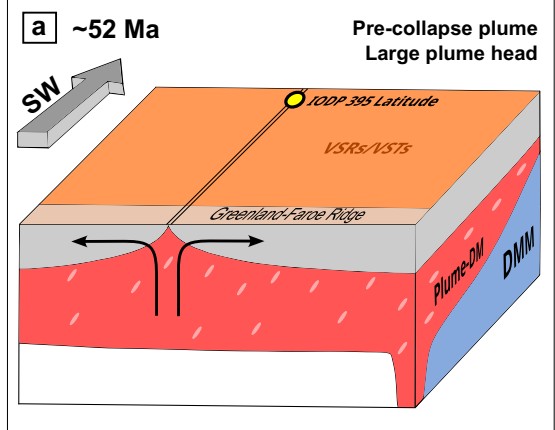
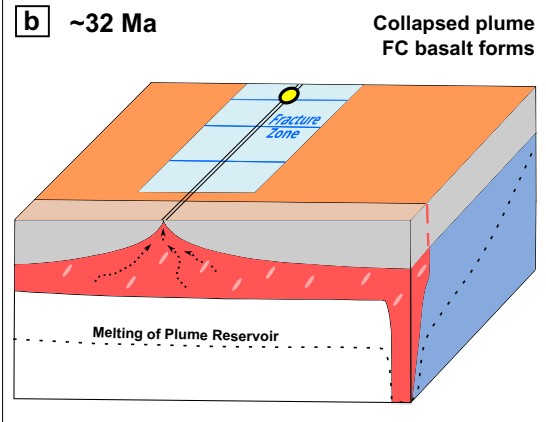
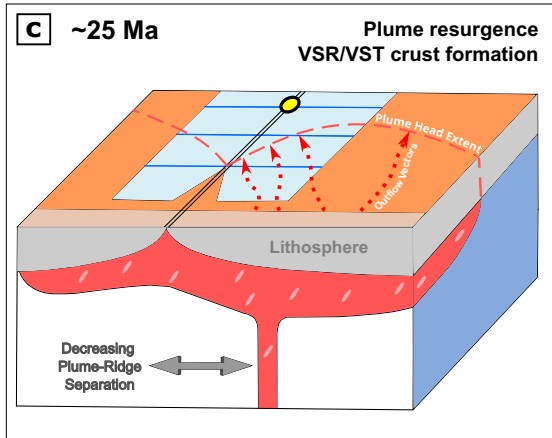
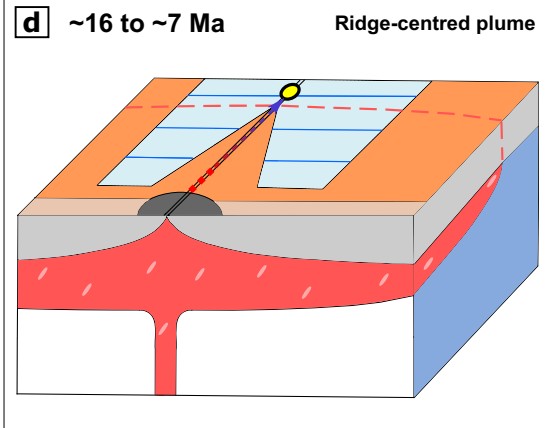

**Fig. 6 | Schematic cartoon of plume collapse and resurgence. a** Decompression melting of a large plume head beneath the North Atlantic rift generates anomalously thick oceanic crust. Orange panel = smooth thick crust; divergent black arrows indicate passive corner flow at the mid-ocean ridge; pink ovals within the red plume = enriched mantle heterogeneities (e.g. pyroxenite). **b** Reservoir of plume material consumed beneath the rift, triggering the collapse of its regional influence. Turquoise panel with blue lines = fractured thin crust. **c** Ridge axis migrates towards the plume conduit, causing resurgence of the Iceland plume. Red dashed line = extent of the plume head during plume-ridge interaction; red arrows indicate oblique outflow of plume material producing limited melting beneath thick lithosphere, but eventually melting to form isotopically enriched basalts at the IODP sampled latitude. **d** Iceland plume conduit becomes ridge-centred at ~16 Ma, triggering extensive melting of upwelling mantle on Iceland[50,51]. Red-to-blue arrow indicates axial outflow, causing progressive exhaustion of enriched mantle components and forming isotopically depleted basalts at the IODP sampled latitude. Figure redrawn with permission from concept in Fig. 1 in Ribe et al.[49].

V-shaped ridges and troughs[43]. At ~38 Ma, transform fault generation began, and a clearly visible boundary between thinner/rougher and thicker/smoother crust receded towards Iceland at a rate of ~88 km/myr[4].

Our geochemical modelling results show that by ~32 Ma, the Iceland plume had no detectable thermal influence at Site FC, whose present-day latitude is ~60°N. Thus, it appears that the composition of basalt glass samples records a dramatic collapse of plume influence. Considering a range of structural and geophysical observations, we infer that this collapse finally occurred at ~38 Ma following a period of plume waning. This inference is consistent with regional sedimentary and palaeoceanographic records[44].

From a fluid dynamical perspective, this long-term behaviour can be accounted for by invoking a head-tail plume structure[14,45]. Waning plume influence likely reflected progressive melting, depletion, and cooling of plume head material beneath the developing rift, whilst collapse at ~38 Ma probably reflects late-stage melting of the plume head that focuses magmatism above the narrow plume conduit[14,45,46]. Widespread cooling results in thinner oceanic crust and facilitates the generation of fractured crust by transform faulting[6,13] (Fig. 6b).

By Neogene times (i.e. after 23 Ma), V-shaped ridges and troughs flank the Reykjanes Ridge, and associated basalts exhibit strong plume influence, indicating that the boundary between fractured and smooth crust represents a resurgence of Iceland plume influence, which grew at a rate of ~40 km/myr (Fig. 1c)[4]. The accompanying increase in mantle potential temperature of ~50 °C, combined with an increase in crustal thickness, may have acted to inhibit transform faulting, initiating the process of their southward elimination (Figs. 1 and 6c)[13,47,48].

Migration of the ridge axis towards the plume centre between 34 and 16 Ma could also explain plume resurgence together with the observed decrease in the pyroxenite contribution to the mantle source between 14 Ma and the present day[13,49] (Fig. 6c, d). When the plume conduit was positioned off-axis between 34 and 16 Ma, plume outflow could have occurred beneath thick lithosphere oblique to the Reykjanes Ridge[50,51]. In this way, decompression melting would have been suppressed until plume flow intersected the ridge axis, thus enabling fusible pyroxenite components to be retained[29] (Fig. 6c). Between 16 and 7 Ma, when the plume conduit was positioned at the ridge axis, outflow could have been focused beneath the Reykjanes Ridge, progressively melting and exhausting the pyroxenite component by the time outflow reached the sampled latitude (Fig. 6d).

An outflow velocity of 40 km/Myr, consistent with the gradient of the smooth/fractured crust boundary, would generate a ~17 Ma delay between outflow from the plume and its sampling by passive isentropic melting along the Reykjanes Ridge 700 km south of the reconstructed plume track[51]. As a result of this delay, the observed decrease in radiogenic isotope enrichment between 14 Ma and the present day

can be linked with migration of the mid-ocean ridge towards the plume conduit between 31 and 17 Ma (Fig. 6c, d).

In conclusion, the IODP drilling results presented here provide significant insights into the long-term development of the Iceland plume. Observed geochemical constraints can be directly applied to interpretations drawn from geophysical[6], structural[10,28] and sedimentological[44] observations. This geochemical evidence requires substantial changes in the degree of plume-ridge interaction over time, highlighting the dynamic, unstable nature of mantle plume upwellings and their influence on magmatic and surface processes[2].

# Methods

## Drilling operations
IODP drilling operations, site descriptions and sample descriptions can be found in the Proceedings of IODP Expedition 395[11]. Whole-rock and glass basalt samples were taken from five principal drill holes (U1555I, U1563B, U1554F, U1562B and U1564F). The first four sites were drilled during IODP Expedition 395C in 2021. Site U1564F was drilled during Expedition 395 in 2023. Samples were selected to avoid alteration and span the complete depth range of basalt recovery within each hole. Each hole cored through over 100 m of basement, with a recovery ranging from 43 to 66%.

## Major and trace element analysis of whole-rock
Whole-rock sample preparation was carried out at the University of Cambridge, UK. Fresh unaltered samples were cut into ~5 cm pieces and dried prior to crushing using a steel jaw crusher and powdering in an agate ball mill. Sample powders were then divided for XRF and ICP-MS analysis. Here, we present whole-rock major and trace element data for hole U1564F at Site FC only, which supplements limited glass availability (see 'Major and trace element analysis of basalt glass').

Major element (and selected trace element) concentrations of whole-rock samples were measured using a Philips PW2404 wavelength-dispersive sequential X-ray fluorescence spectrometer (XRF) at the University of Edinburgh. These measurements are not directly exploited in this study but are provided for reference purposes in the Source Data file. A detailed description of the sample preparation procedures is provided by Passmore et al.[52]. Fused glass discs and pressed pellets were prepared from powdered samples. Major and trace elements were analysed using fused glass discs and pressed pellets, respectively, following the analytical procedures of Fitton et al.[53].

Accuracy and precision of XRF analysis were determined by repeat measurements of USGS international standard materials BIR-1 and BE-N. In both cases, the accuracy of major element measurements is better than 1%, with the exception of $Na_2O$, which has an accuracy of ≤5%. Precision is typically better than 1%. These data are consistent with previously published XRF analyses on basaltic samples[52].

For ICP-MS analysis, 0.1 g of powder was digested in concentrated $HF-HNO_3$ overnight on a 115 °C hot block. Fluorides were redissolved with concentrated $HNO_3$ and MilliQ $H_2O$, before dilution to 50 ml. All samples were analysed on a Perkin Elmer Nexion 350D ICP-MS at the University of Cambridge, UK. Calibration standards were BIR-1, BHVO-2 and BCR-2. Internal standards were 10 ppb Rh, In and Re, and each sample was prepared in 1% $HNO_3$. Instrumental drift was less than 10%. ICP-MS sensitivity was $5 \times 10^5$ cps/ppb In, with CeO/Ce ratios approximately 2%. Raw intensities were blank-subtracted and internal standard normalised before calibration calculations were performed. REE element signal intensities were corrected for any oxide overlaps using correction factors that had been previously generated.

Accuracy and precision of ICP-MS measurements were monitored by repeat measurements of external USGS reference materials BCR-2, BHVO-2 and BIR-1 throughout the analytical sessions. Across all standards, the accuracy of REE measurements was better than 3% and typically better than 2%. Precision was also better than 3%. Full

accuracy and precision statistics of standard analyses are provided in the Source Data file.

## Major and trace element analysis of basalt glass
Glass chips of size 1–2 mm were picked from crushes of selected samples. The glass chips were cleaned with MilliQ $H_2O$ and inspected with a reflected light microscope to avoid microlites. Three to five chips per sample were set into epoxy resin mounts.

For major element analysis, a Cameca SX100 EPMA at the University of Cambridge was used. Major element oxides were measured using five wavelength-dispersive spectrometers with counting times on elements ranging from 10 to 60 s. Beam conditions of 15 kV accelerating voltage, 10 nA, and a spot size of 10 μm were employed. Up to 25 total analyses per sample were completed (5 spots across up to 5 chips, depending on chip quality/availability). The accuracy and precision of measurements were monitored by regular measurement of external standard glasses VG-2 and NMNH 113716-1 from the Smithsonian Museum of Natural History. Accuracy for all major element oxides of VG-2 was ≤5% and was better than 2% for MgO and FeO, which are the only major element oxides used in this study. Accuracy for NMNH 113716-1 was ≤5% for all major element oxides except for $Na_2O$ and $P_2O_5$. Accuracy was ≤3.5% for $SiO_2$, MgO and FeO. Precision for both standards was ≤3% (1 standard deviation) for all elements not close to the detection limit. Significant analytical drift was not observed. Overall, this performance is comparable with other published studies that analysed these standards (see Source Data) as well as with studies employing EPMA for natural glasses[52]. It was deemed adequate for the purposes of our study. Uncertainty is not provided for these reference standard values. However, variations in reported values from a number of published studies are ≤3% for VG-2 and up to 7% for NMNH 113716-1. In particular, studies repeatedly report $Na_2O$ values for NMNH 113716-1 that are more than 5% higher than the accepted value (see Source Data).

Major element data that have been corrected against the accepted composition of VG-2 are also provided in the Source Data file, together with the factors applied for correction. This correction improves the accuracy of major element oxides for the NMNH 113716-1 standard to ≤2% for all measurements above 0.5 wt% with the exception of $Na_2O$, which, as stated above, does not have a reliable accepted concentration. This correction procedure follows the method of Helz[54] and is similar to the inter-laboratory bias corrections applied by Gale et al.[15] for the VG-2 glass standard.

Glass chips were included in the sample mean if the standard deviations across the chips for selected major element oxides did not exceed specified thresholds (i.e. $Na_2O < 0.1$, $Al_2O_3 < 1$, $SiO_2 < 0.5$, CaO < 0.5, MgO < 0.5 and total oxides < 1.5 wt%). Variation between chips of other major element oxides was manually inspected after filtering to ensure that clean glass had been targeted. The number of chips for each sample that passed filtering conditions is included in the Source Data file, in addition to the average standard deviation of oxides across each chip and the chips included in the sample mean. Raw data are also provided to enable readers to apply their own filtering conditions.

For trace element analysis, an ESI NWR193 laser system coupled to a Perkin Elmer Nexion 350D Inductively Coupled Plasma Mass Spectrometer (LA-ICP-MS) at the University of Cambridge was employed. This technique requires a 50 μm diameter laser beam, a laser repetition rate of 20 Hz and a laser fluence of 4 J/cm² in order to ensure optimum signal intensity while minimising downhole fractionation. LA-ICP-MS data acquisition settings were 1 sweep per reading, 80 readings, 1 replicate and total data acquisition lasted 60 seconds (i.e. approximately 1 data point for each element per second) with a laser warm-up time of 20 s for each spot analysis. The ICP-MS dwell time for each mass was dependent upon the isotope and concentration

of the element in the samples, but was typically 20–40 ms for trace elements.

Up to 15 individual analyses were carried out per sample (3 spots across up to 5 chips, depending upon chip quality/availability). At least 8 samples were analysed per drill hole, covering the full depth interval with the exception of hole U1564F, where only two glass samples were of sufficient quality to obtain reliable data. The GLITTER (v4.5) software package[55], developed by GEMOC (https://www.glitter.mq.edu.au), was used for data reduction, including background subtraction, drift correction and external calibrations. Several external glass reference samples were also analysed to ensure data quality, including NIST-610, NIST-612, NIST-614, BIR-1G, BCR-2G and BHVO-2G. The external calibration standard was BHVO-2G. Both NIST-612 and BHVO-2G were evaluated as calibration standards, but BHVO-2G was preferred for the matrix-matching composition. LA-ICP-MS raw intensity drift during an analytical session of 8 hours is typically less than 10%, based upon raw counts for NIST standards and compensated for by the internal standard calculations in the GLITTER Software; no other drift corrections are used. Silica concentration for BHVO-2G from the GeoReM database was employed as the normalising internal standard.

Accuracy and precision of LA-ICP-MS analysis were monitored by repeat measurements of external reference samples BIR-1G and BCR-2G. Accuracy for BIR-1G is ≤5% for most trace elements and for all REEs, which are the primary focus of this study. Accuracy for BCR-2G for REEs is ≤5%, except for Ce, which is 6%. Most REEs on both standards have an accuracy ≤3%. Precision of trace element analyses was variable but better than 10 % for most REEs and trace elements of sufficient concentration above the detection limit. Details of the accuracy and precision of these standards are provided in Source Data.

Glass chips were included in the sample mean if the standard deviations across chips for selected trace elements did not exceed specified thresholds (i.e. La < 0.5, Ce < 0.9, Nd < 1 and Pb < 0.5, Li < 0.6 and Sr < 9 ppm). Variation of other trace elements between chips was manually inspected after filtering to ensure that clean glass had been targeted. The number of chips for each sample that passed filtering is included in the Source Data file, in addition to the average standard deviation of trace elements across each chip and the chips included in the sample mean. Raw data are also provided to enable readers to apply their preferred filtering conditions.

$SiO_2$ concentrations of glass chips measured by EPMA are used to normalise the trace element data measured by LA-ICP-MS. This correction is proportional, such that a 1% error in $SiO_2$ yields a 1% error in trace element concentration. EPMA analysis achieved an uncorrected accuracy of < 2%, and therefore, the propagated uncertainty in trace elements is < 2%. This uncertainty is typically smaller than the LA-ICP-MS measurement uncertainty. Trace element data calculated using raw $SiO_2$ concentrations, as well as trace element data calculated using $SiO_2$ concentrations corrected on the VG-2 glass standard, are provided in the Source Data file. The principal component analysis and modelling results presented in this study exploit the corrected trace element data. However, identical analysis using uncorrected measurements yields almost identical conclusions. For example, using uncorrected glass measurements from Site FC yields an optimal mantle potential temperature that is only 3 °C higher, which is within the accepted temperature range used to match observations.

### Nd radiogenic isotope ($^{143}Nd/^{144}Nd$) analyses

Weighed whole-rock sample powders were dried at 95 °C before redissolving for ion exchange chromatography. Nd was purified from these samples using a two-stage column procedure: Eichrom TRU spec columns isolated the REE fraction, before Eichrom Ln spec columns purified Nd. Samples were aliquoted and diluted with 2% $HNO_3$ to equal Nd concentrations for measurement. $^{143}Nd/^{144}Nd$ ratios were then determined on a Thermo Scientific NEPTUNE Plus multi-collector ICP-MS. Sample preparation and measurement for samples

from holes U1555I, U1563B, U1554F and U1562B (VST-1–VSR-3) were carried out during July-August 2023. Sample preparation and measurement for site U1564F (FC) were carried out during October–December 2024.

In 2023, the JNdi-1 (20 ppb) Nd bracketing standard was determined at $^{143}Nd/^{144}Nd$ of 0.512001 ± 8 (2 s.d., $n = 16$), and all data are normalised to JNdi-1 $^{143}Nd/^{144}Nd$ of 0.512115[56]. To ensure reproducibility of the radiogenic isotope ratio analyses, USGS rock reference materials BCR-2 and BHVO-2 were taken from powders to Nd fraction and analysed simultaneously with the samples. These standards were measured with $^{143}Nd/^{144}Nd$ of 0.512641 ± 13 (2 s.d., $n = 4$) for BCR-2 and $^{143}Nd/^{144}Nd$ of 0.512993 ± 15 (2 s.d., $n = 4$) for BHVO-2. These measurements compare to accepted GeoReM values of 0.512635 ± 29 (1 s.d.) and of 0.512979 ± 14 (1 s.d.) for BCR-2 and BHVO-2, respectively, implying good external reproducibility. All procedural blanks contained negligible Nd.

In 2024, the JNdi-1 (20 ppb) Nd bracketing standard was determined at $^{143}Nd/^{144}Nd$ of 0.512117 ± 2 (2 s.d., $n = 15$), and all data are normalised to JNdi-1 $^{143}Nd/^{144}Nd$ of 0.512115[56]. BCR-2 and BHVO-2 USGS standards were measured with $^{143}Nd/^{144}Nd$ of 0.512633 ± 7 ($n = 3$) for BCR-2 and $^{143}Nd/^{144}Nd$ of 0.512985 ± 4 ($n = 2$) for BHVO-2. All procedural blanks and blanks that had undergone column chemistry contained negligible Nd. Five samples (including a USGS standard) from the 2023 run were reanalysed in the 2024 run. These samples recorded equivalent results within measurement error. In addition, two samples from the 2023 run were reprocessed through the entire chemistry. These samples returned the same results within measurement error as in 2023.

Basalts from hole U1564F are extensively altered[11]. Sea-water alteration has a negligible effect upon the $^{143}Nd/^{144}Nd$ of oceanic basalts[57]. However, for verification, acid pre-leaching was carried out on a second batch of U1564F samples prior to digestion and column chemistry. Acid leaching followed the procedure of Weis et al.[57]. Two leached residues were taken for column chemistry and $^{143}Nd/^{144}Nd$ analyses. Measured $^{143}Nd/^{144}Nd$ ratios for leached and un-leached basalt powders were equivalent within measurement error, suggesting Nd isotopes are unaffected by even severe alteration.

A minor age correction was applied to all isotopic measurements. For that sample closest to the sediment/basement interface, age is constrained by a magnetic anomaly, which is then verified using palaeotological biomarkers at the base of the sediment pile. The ages are 2.8 Ma for U1555I, 5.2 Ma for U1562B, 12.7 Ma for U1554F, 13.9 Ma for U1562B, and 32.4 Ma for U1564F. In this study, it is assumed that there is no significant age variation within the basalt drilled section. Due to the long-half life of the Sm-Nd decay system, small variations in down-hole age will not quantitatively affect the results.

### Principal component analysis

REE measurements from all sites were combined with previously published datasets from axial dredges. The compiled studies consist of Gale et al.[15], Murton et al.[16] and Jones et al.[17]. Duplicate analyses were removed. Dredge samples less than 60 km south of the southern transform fault of the CGFZ were removed to mitigate the effects of thermal cooling caused by the presence of the transform fault[25]. Data sources are provided in Source Data. Dredge materials have inherently greater spatial uncertainty compared to ocean drilling. This uncertainty is partly addressed by the high axial sample density combined with the fact that axial geochemical trends in trace element and radiogenic isotope composition are well characterised[15–17,30].

Whole-rock samples from Site FC were filtered for plagioclase accumulation since large phenocrysts were observed in some thin sections[11]. Sr/Y is an appropriate proxy for determining the extent of plagioclase crystal accumulation because Sr is highly compatible in anorthitic plagioclase compared with Y[58]. Plagioclase accumulation is also expected to elevate $Al_2O_3$ content and may generate an Eu

anomaly. PCA was carried out on whole-rock samples from Site FC using the method as described below. $P_2$ was found to dominantly reflect plagioclase accumulation with correlated loadings on Sr, $Al_2O_3$ and Eu. Whole-rock measurements were removed if Sr/Y > 3.2 (glass average = 2.7), which correlates with $Al_2O_3$ and generally gives a $P_2$ score elevated with respect to the glass samples.

The combined REE dataset was normalised by subtracting the mean and dividing by the standard deviation for each element. Samples for which all REEs were not reported were omitted. PCA was implemented using the decomposition module and PCA algorithm in Scikit-learn (v1.5.1). Elemental loadings were calculated from the component matrix calculated, and principal component scores were computed using matrix multiplication of the normalised data with the loading matrix. PCA element loadings and principal component scores are provided in the Source Data file.

The effect of melt degree and fractional crystallisation was projected into PCA space with the aid of the pyMelt library (Fig. 2c). To examine the effect of temperature anomalies, a forward-modelled composition and parameterisation based upon the glass composition at Site FC was created, and mantle potential temperature was incrementally increased. To examine the effect of fractional crystallisation, a forward-modelled composition and parameterisation that represents the central cluster of samples within the Icelandic chemical gradient was created, and olivine was crystallised out using an in-built pyMelt function. Both processes largely resulted in uniform enrichment/depletion of REE concentrations. The calculated melt compositions were then analysed using PCA and found to plot predominantly along the trajectory of the Gd loading vector. To improve interpretability and to ensure these processes resulted in a horizontal array, all PC scores and loadings were rotated to align the Gd vector with the positive horizontal axis. These rotated axes are referred to as $P_{1r}$ and $P_{2r}$.

## Geochemical modelling

First, measured REE concentrations are corrected for crystallisation of olivine, following the method of Tatsumi (1983)[59]. We assume a Fo number of olivine that is in equilibrium with the mantle of 90[60], a Kd (Fe/Mg) = 0.3[61], and a constant ratio $Fe^{3+}/\sum Fe = 0.15$, consistent with values from the Reykjanes Ridge[62]. In this way, we calculate the amount of olivine that would need to be added to the basalt liquid composition until olivine, which is in equilibrium with the rock, reaches an Fo of 90. We then calculate the original trace element concentration of the magma by diluting it in proportion to the volume of added olivine. Samples from all sites are corrected in this way to enable quantitative inter-site comparison.

pyMelt (v2.3) was used for REE forward modelling. We parametrised the mantle source using a combination of the hydrous lherzolite lithology of Katz et al.[39] and silica-undersaturated KG-1 pyroxenite. The trace element composition of the lherzolite was at first assigned that of Workman and Hart[40]. For VSR/VST sites, it was not possible to obtain acceptable fits using this source composition (Fig. 5). Therefore, a revised mantle source composition was calculated by calibration, which provides a closer approximation to the depleted component of the Iceland plume.

This source was obtained by matching both the observed REE composition of dredge samples where VSR-1 intersects the mid-ocean ridge and the observed crustal thickness of 8.6 km at this location (White et al.[19]). The composition of this calculated mantle source is provided in the Source Data file. The trace element composition of KG-1 was produced by mixing depleted mantle (Workman and Hart[40]) and subducted oceanic crust (Stracke et al.[63]) in a 1:1 ratio[64]. Partition coefficients of Gibson and Geist[65], were assumed. Mineralogy of the lherzolite was specified as defined for spinel peridotite by McKenzie and O'Nions[38], and for the pyroxenite as defined for KG-1 by Matthews et al.[37]. Exhaustion of clinopyroxene during melting was set at 15%. The pyMelt default parameterisation of the garnet-spinel mineral transition

was used, including a linear function in temperature-pressure space with a gradient of 1/666.7 and an intercept pressure of 400/666.7 for garnet-out, and 1/666.7 and 533/666.7 for spinel-in[38]. The spinel-garnet transition was translated to a lower pressure for pyroxenite melting to account for greater garnet stability.

The isotopic composition of the homogenised melt was calculated by applying a simple binary mixing model between the lherzolite and pyroxenite melts (Supplementary Fig. 1). The Nd isotope ratio of the individual melts is inherited from the isotopic composition of the mantle sources. A range of $^{143}Nd/^{144}Nd$ contents for pyroxenite and for the enriched Icelandic end-member exist (e.g. 0.5127[33], ≤0.5128[34] or 0.5129[32]). The value used in this study is 0.5127. A higher value of 0.513156 ($\varepsilon_{Nd}$ = 10.1) was picked for the lherzolite component, which matches the isotopic composition of Reykjanes Ridge dredges south of 61.2°N. This value is similar to the value of 0.51320 used for the peridotite end-member by Koornneef et al.[33]. It is also similar to the value of 0.51313 estimated for average DMM by Workman and Hart[40].

For each site, the lithological contribution of pyroxenite and lherzolite components was calculated using pyMelt. Mantle proportions were varied to ensure that lithological contributions from pyroxenite matched the proportion required by the simple isotopic mixing model (Supplementary Fig. 1). Crustal thickness was calculated within pyMelt by finding the depth at which the pressure of produced overlying melt of crustal density (2.9 g/cm³) is equal to the pressure of melting. The base of the crust was taken as the top of the melting column. Instantaneous fractional melts produced beneath this depth were integrated assuming a triangular melting regime for each lithology. The trace element composition of the erupted melt is then calculated by homogenising the pyroxenite- and lherzolite-derived melts, weighted by the relative contribution of each lithology to the pooled melt/igneous crust. Mantle water content was set at 0.01 wt% for Site FC, consistent with reference estimations for DMM[66]. For the axial VST-1, VST-1, VSR-2 and VSR-3, water content was set at 0.015, 0.016, 0.016 and 0.014 wt%, respectively. These values were chosen using $H_2O$/Ce values that match axial dredge samples with similar degrees of mantle enrichment ($\varepsilon_{Nd}$).

## Testing model parameters

We carried out temperature sensitivity analysis by varying the potential temperature on either side of the optimal value. In this way, a range of temperatures that reflects measurement uncertainty was obtained (Fig. 4). Significantly, the range of temperatures for Site FC does not overlap those of the VSR/VST sites. Geochemical modelling is dependent upon the estimated melt productivity of the mantle source for a given mantle potential temperature. Melt productivity is dependent upon the choice of mantle source parameterisation together with the mantle water content. The latter was found to have a very minor effect on melt generation and melt composition: trebling water content for melting hydrous lherzolite from 0.01 wt% to 0.03 wt% increases the predicted crustal thickness by 0.2 km where forward-modelled light REE concentrations are still matched within the degree of uncertainty.

The conclusions of this study depend upon the relative difference in the depth and degree of melting calculated for different sites. It is therefore important to assume a self-consistent set of assumptions regarding mantle source parameterisation and mineralogy. Uniform changes to these assumptions do not affect the relative differences and the conclusions that we make. As a test, we changed the lherzolite melting parameterisation to that constructed by Ball et al.[67]. Their lherzolite parameterisation generates less overall melt for a given mantle potential temperature compared to Katz et al.[39], yielding a more REE-enriched aggregated melt composition. At Site FC, this model test required a mantle potential temperature of 1322 ± 15 °C, which is, within the range of petrological uncertainty, essentially the ambient value. The cumulative melt thickness represents a crustal

thickness of 6.0 km, compared with our original value of 5.8 km. This test demonstrates that our conclusions are robust to reasonable changes in mantle source parameterisation.

The effect that including variable (but small) amounts of pyroxenite has upon melt production and composition also depends upon modelling assumptions. Changing the $^{143}Nd/^{144}Nd$ value of the pyroxenite component from 0.5127[33] to 0.5129[32] at Site FC increases the predicted value of $\varepsilon_{Nd}$ for the homogenised melt by 0.13, which is smaller than the typical measurement uncertainty for $\varepsilon_{Nd}$. At VSR-3 (i.e. hole U1562B), which has a greater expected proportion of mantle pyroxenite (i.e. 2.6%), this change increases the $\varepsilon_{Nd}$ value of the homogenised melt by 0.63. Mantle pyroxenite would be required to increase up to 4.5% in order to match the observed value of $\varepsilon_{Nd}$. The consequent enrichment of REEs within the melt that this additional pyroxenite requires can be offset by a -15 °C increase in mantle potential temperature. This temperature increase does not materially alter the conclusions of our study. On the contrary, it augments the relative difference in temperature predicted for the FC and VSR/VST sites.

Finally, an alternative approach for parameterisation of the garnet-spinel transition assumes that transition depths are isobaric and uses the pressure range constrained by the thermodynamic framework of Tomlinson and Holland[68] which are 2.23–2.52 GPa (i.e. 69–78 km). Testing these values led to a very minor change in calculated heavy REE concentrations, requiring no change to the predicted mantle potential temperature for Site FC. At higher temperatures (e.g. VSR-3: hole U1562B), a minor increase in the proportion of melting within the garnet stability field leads to depletion of heavy REEs whilst light REEs remain largely unchanged. To improve model fit, one option is to decrease mantle potential temperature whilst slightly lowering the mass fraction of mantle pyroxenite. The revised potential temperature is -1400 °C, which remains significantly greater than ambient asthenospheric temperature and the predicted temperature for Site FC.

## Data availability
The data generated in this study are provided in the Source Data file and are available in the Figshare repository at https://doi.org/10.6084/m9.figshare.31488925. Gravity data are the Sandwell and Smith Gravity grid and can be found at the following https://doi.org/10.1126/science.1258213[69]. Bathymetry data are the GEBCO Compilation Group (2025) GEBCO 2025 Grid (https://doi.org/10.5285/37c52e96-24ea-67ce-e063-7086abc05f29)[70]. Source data are provided with this paper.

## Code availability
All computer code necessary to reproduce the results of this study is publicly available. Information regarding modules and algorithms implemented can be found in the 'Methods' section.

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

## Acknowledgements

This research used samples and other measurements provided by the International Ocean Discovery Program (IODP). Funding for this research was provided by UKRI/National Environmental Research Council (NERC NE/W007150/1 awarded to N.W.). We are grateful to the crew, technical staff and scientists of Expedition 395 onboard the *JOIDES Resolution* and to repository curation staff at Bremen and Texas. We thank M.-L. Bagard, I. Buisman, G. Calder, G. Fitton, M. Tarique and E. Tipper for laboratory analysis.

## Author contributions

C.P. carried out data acquisition, interpretation, modelling and drafted the manuscript. C.Y.T. contributed to data acquisition and interpretation. N.W. conceived the study, provided financial and other resources for data acquisition, contributed to interpretation and revised the manuscript. J.M. contributed to data acquisition, data processing and interpretation. B.M. contributed to the interpretation. S.G. and J.D. provided laboratory resources. R.P.T. was co-chief scientist of IODP Expedition 395. All authors contributed to the editing of the manuscript. Members of the IODP Expedition 395 Science Party contributed to sample collection and description, discussion and editing of the manuscript.

## Competing interests

The authors declare no competing interests.
