## [Transparent Peer Review file · Nature Communications]

Collapse and resurgence of the Iceland mantle plume

Corresponding Author: Mr Callum Pearman

Version 0:

Reviewer comments:

Reviewer #1

(Remarks to the Author)

This study uses the geochemistry of basalts from five drilling sites in the North Atlantic to study the evolution of the Iceland plume and how it influences seafloor spreading. The authors argue that the Iceland plume influence collapsed rapidly at 38 Ma and was then progressively re-established. This is an impressive study, and the authors provide high quality geochemical data and geochemical modelling to support their ideas.

Comments

1. The authors argue that changes in the mantle source composition are needed to explain the different trace element patterns in FC samples compared to VST/VSR samples. They show in Figure A3 how the trace element pattern changes between different mantle sources (i.e., DM from Workman and Hart and calibrated plume DM) if melting conditions are kept the same. But I am not convinced that the trace element differences cannot be generated by different degrees of melting. Further discussion and visualization of how the trace element pattern changes if melting conditions are changed for a constant mantle source could clarify this.

2. Both FC samples and SGFZ samples are erupted close to transform faults. In lines 288-289 the authors note "In addition, dredge samples immediately south of the GFZ were removed to avoid atypical melting conditions affected by the transform fault." Could the differences in trace element geochemistry between FC samples and VST/VSR samples be related to different crustal environments? The only samples that have similar trace element geochemistry to FC samples are the SGFZ samples that also erupted close to a transform fault.

3. Some of the data references for previously published data are not clear. For example, data references for both trace elements and isotopic compositions for SGFZ samples are unclear. Also, the authors state that they use DMM composition from Workman and Hart (2005), but it is not clear which DMM composition (i.e., average DMM, enriched DMM or depleted DMM).

Figure comments

Figure 2

Panel a – The authors state in lines 43-44 that samples south of 61.2°N are depleted relative to N-MORB and D-MORB. Adding N-MORB and D-MORB compositions to Figure 2a would help the reader visualize this. VST-2 is labelled twice in panel 2a both in light blue and dark blue. Light blue label is probably incorrect.

Panel c – Plotting the lowest distance on top of higher distances might help this figure. Some of the samples along the ridge show relatively low distances but they are overlain by samples with higher distances.

Figure 3

The most enriched seamounts on the ridge-axis are not shown in Figure 3. The $^{143}\text{Nd}/^{144}\text{Nd}$ data from the enriched VST-2 seamount is still shown in this plot but this locality is excluded from the trace element modelling because of enrichment in trace element composition. This biases the $^{143}\text{Nd}/^{144}\text{Nd}$ of the ridge-axis dredges towards more homogenous depleted $^{143}\text{Nd}/^{144}\text{Nd}$ and ignores small-scale heterogeneity in the mantle source along the Mid-Atlantic ridge. While the full variability of the VST/VSR $^{143}\text{Nd}/^{144}\text{Nd}$ data is shown.

Figure 4

The classification of VST-1 samples into on axis VST-1 and VST-1 confused me because it has not been used previously in the paper. This made me wonder if on axis VST-1 and VST-1 plot in the same space on the PCA plot, and if they have

similar $^{143}\text{Nd}/^{144}\text{Nd}$ composition. It would be good to see this classification also in Figures 2 and 3. Also, on axis VST-1 have a very similar color to VST-2 in Figures 2 and 3, making it is easy to confuse the two groups together.

Reviewer #2

(Remarks to the Author)

Review of 'Ocean Drilling Reveals Collapse and Resurgence of the Iceland Mantle Plume' by Pearman et al.

This paper explores an interesting question regarding the temporal continuity of the Iceland Mantle Plume using an important new dataset obtained via IODP cruise 395. Given the paper's ability to link this geochemical data to some of the down-core geophysical observations, it is a useful study with a potentially impactful discussion and therefore should be published. However, I believe there are a number of items that need to be addressed prior to publication so I recommend revisions.

When looking at the dataset provided as supplementary material I have noticed a few issues with the analyses. I am slightly concerned that some of the standard reference material values are outside what is deemed acceptable. The authors should look into this in more detail and investigate why this might be. Also, with the EMPA analysis, the dataset provided is lacking detail I would expect (number of analyses for each sample, standard deviation) and, if the authors are willing, I would advise them to publish the raw dataset in a separate sheet for openness and use in the future by readers.

With the melt modelling (section 2.3 Geochemical modelling of Fractured Crust), you have clearly stated your input parameters in the main text (and expanded on them in the methods) which is great. However, there is no discussion of uncertainties in those and how your results would change if you varied them slightly. Please add a section (into the main text and/or methods) on the sensitivity of the model and the resulting uncertainty if the input parameters are varied slightly.

I believe that a few more markers for the readers towards figures, methods section, appendixes, supplementary data, will make it a lot clearer and easier for readers to follow, and may help them realise there is a dataset they could utilize (leading to more citations).

Those are just a couple of my main concerns that I feel must be addressed before publication. Below gives further comments and expands (giving guidance) to the comments above.

Overall, I enjoyed reading the manuscript – it is well written overall – and contains some interesting discussion that will be used both in the Icelandic mantle field but hopefully also elsewhere on similar sites. I hope the authors find my comments helpful and I hope to see the manuscript published soon. Emma J. Watts.

Abstract:

1. In the abstract add in the age after 'continental breakup' in sentence #4. Now, it isn't clear the timelines that this has occurred over.
2. Please consider splitting sentence #5 up as its meaning and impact gets a little lost given the length and complexity of the sentence.

Main Text

Line 4: While I agree with the concept and reference used. The number of deep mantle plumes is not widely agreed, so I would suggest amending the sentence so that you rather show that it was a recent study [2] suggested that there were 18 deep mantle plumes.

Line 12: change 'may' to 'are suspected' as the use of may sounds like you will put an alternative hypothesis after but there is none stated.

Line 13: The sentence with the list of boreholes and ages is not clear. Please clarify what the ages are of? Are they the deepest depth of rock found in the borehole, the age of the VSR or something different? This should be made very clear to all readers.

Line 17: Is FC the name of the borehole? Try to clarify this please.

Line 24: Add in rough age of Neogene to aid readers.

Line 27: Please state that you're using radiogenic isotopes and expand or give examples on what you mean by contextual geophysical observations.

Line 32: Please include a date of continental break-up for context.

Line 35: Could you be more specific about when the plumes influence re-established?

Line 40: Why was whole rock ICP-MS only done on the FC sample? This is not explained or justified in the text. If you can't fit an explanation in the main paper for word limit reasons, please include in the methods section and point the readers to that section.

Line 41: It's great that you have used previously published REE data from dredged samples to be able to make further conclusions. However, there is no discussion of that data in the methods section. Please amend this and add in a section on that data, and the limitations of it given that it is dredged data and will therefore have more of a spatial uncertainty etc. Given this study looks into the spatial and temporal changes of the Iceland plume, the (age and location) uncertainties in all data should be acknowledged and discussed.

Line 45: Please point the reader to Figure 2 when discussing the GFZ.

Line 61: Lower than what? Please be clear in what you are comparing to.

Line 78: Add in the words previously and radiogenic to make the sentence read: 'Along the Reykjanes Ridge, the observed gradient in radiogenic isotopic enrichment has previously been interpreted as progressive dilution of an enriched plume source by mixing with depleted mantle during southward outflow from Iceland [23] (Fig. 3a).'

Lines 87-92: This is very interesting to see the variation with time. I think it would be even more helpful to the community if you could also assess the $^{87}\text{Sr}/^{86}\text{Sr}$ or $^{206}\text{Pb}/^{204}\text{Pb}$ with time. This would also then allow to look in isotopic-isotopic space e.g. $^{87}\text{Sr}/^{86}\text{Sr}$ vs $^{143}\text{Nd}/^{144}\text{Nd}$ and get a better grasp of the mantle end-members contributing and allow for easier discussion of the pyroxenites etc. I understand that would require a lot of work, but I think it could significantly help the study go to the next level of discussion.

Lines 106-108: When stating you think that the REE compositions and crustal thickness provide a more robust measure of

plume influence - what statistics are used to back this up with? If this is a sweeping statement without any method of it being a 'measure' please amend the sentence. If it can be used as a mathematical predictor this needs to be expanded on and explained, including the limitations.

Line 117: How much is trace pyroxenite? For reproducibility, please state a numerical value.

Line 134: If you believe that VST-2 drilled a seamount, why was it included in the first place?

Line 136: While I agree some of the variability aligns along the Dy axis on the PCA, however it appears the trend doesn't fully line up and so could be influenced by Gd and Lu too (especially the cream Reykjanes Ridge samples). It might be worth looking at the best fitting linear correlation of the trend (s) for the whole dataset and the individual groups to see if they agree with your statement and if they're statistically significant. This will make your statements more robust.

Figures:

Figure 1

- Is the scale for gravity anomaly the same for both (a) and (b). If so, consider centering across the whole figure so it is below both (a) and (b). This would make it clearer to readers that the scale is the same. If it's not the same, please include a new scale for (b).
- In subfigure (a) The purple lines are quite hard to see on the red and blue background. Please see if there is a way to make them clearer.

Figure 2:

- With the Charlie-Gibbs Fracture Zone – please be consistent with the acronym used to refer to it by. In the main text (Line 45) you state it is GFZ, then in the legend you use CGFZ.
- Please include error bars on your plots of if the error is smaller than the data points state that in the caption.
- When using different colours for datapoints please also consider using different shapes as it may help if people are colourblind or print in black and white.
- In Fig. 2c please make it clearer exactly where the GFZ and BFZ are. At the moment, it is a little ambiguous.

Figure 4:

- Please label the grey box with crustal thickness (I'm guessing) to signify its meaning.

Figure 5:

- Please consider adding what the yellow dot represents to the figure itself as well as the caption. The same applies to the green arrows.

Methods:

Section 5.2.1: When discussing the Nd isotopes please write the isotope ratio out in full as not all readers will know the pair. Also, paragraph 1 in this section (5.2.1) jumps around a bit making it hard for the reader to follow. Please reorder and keep procedures more logical and in order of actual steps. Consider discussing trace element analysis first, then move on to $^{143}\text{Nd}/^{144}\text{Nd}$ analysis.

With the provided data spreadsheet. Please include your standard deviation of your measurements for the standards e.g. BIR-1 as well as the range or standard deviation of the accepted values so that they can be compared. Currently your SiO_2 , MgO and CaO values of BIR-1 appear to be slightly outside of the accepted value stated.

Line 218: What type of beads were made for XRF? Was it lithium tetraborate or something different? What ratio was used of that to your sample? More detail is required for the results to be reproducible.

Line 233: Please add in the accepted values for the rock reference materials BCR-2 and BHVO-2 and the appropriate reference.

Line 238: While I understand you are following the sample writing pattern for other analyses but stating a standard deviation when your $n=2$ is statistically meaningless. Please amend and just write both values.

Line 248: Please consider putting 6M in brackets next to your HCl concentration to aid readers.

Line 270: Please add a reference for the Glitter software used.

Section 5.2.2: When discussing the EMPA and LA-ICP-MS analysis it is clear you have done multiple analyses per sample which is good however more detail in the data spreadsheet is needed. I understand you have summarized the data putting the average major and trace element compositions for each sample on one line however those do not show the number of spots for each nor the standard deviation. This should be the minimum detail included. Ideally you would have this and, on another sheet, you would have the raw data provided so that it may be used by others and is completely transparent. Also, it is not clear when the standards for any of the analysis were carried out. Ideally, they would have been spaced throughout the run with the minimum being at the start and end of the run to check for analytical drift. Please provide more information on this.

Finally, your EMPA VG-2 standard values for SiO_2 , Al_2O_3 and TiO_2 and NMNH 113716-1 values for SiO_2 , Na_2O and TiO_2 are outside of the accepted values. Please go back and investigate this as it is not mentioned at all in either the methods or paper but it is very important as it puts doubts on your values of those elements.

Line 280: Please add a space between 'position.PCA'

Line 281: Please add a table(s) in of the PCA element loadings and principle component cores to the appendix.

Line 283: You discuss the filtering with Sr and Eu but is that the only filter condition you used? If not, please state all filtering requirements.

Line 288: Please clarify those samples were removed. State what the $\text{Sr}/\text{Y} > 3$ and negative P2 mean for the readers.

Version 1:

Reviewer comments:

Reviewer #1

(Remarks to the Author)

I have no more comments on the manuscript and think that it is ready to be accepted.

Reviewer #2

(Remarks to the Author)

The authors have thoroughly responded and addressed the issues brought up during the first review. The manuscript has severely benefited from the edits. The authors have undertaken further analyses (sensitivity analyses) to investigate the uncertainties and the sensitivity of the model and have discussed this within the methods section. Overall it seems the model's outputs don't vary with these uncertainties so I am satisfied with this. The authors have also considered my comment on the standards and have been more open with their data. Given my concerns with the EMPA data they have now applied a correction factor which has improved the standard data relative to accepted values. This has been made clear in the methods section.

I am happy to see the paper in its improved state and look forward to see it published.

Responses to Reviewers' Comments for Manuscript NCOMMS-25-76591

Ocean Drilling Reveals Collapse and Resurgence of the Iceland Mantle Plume

Submission to

by

Callum Pearman, Chia-Yu Tien, Nicky White, John Maclellan, Bramley Murton, Sally
Gibson, Jason Day, Ross Parnell-Turner and IODP Expedition 395 Science Party

Authors' Response to Reviewer 1

General Comments. This study uses the geochemistry of basalts from five drilling sites in the North Atlantic to study the evolution of the Iceland plume and how it influences seafloor spreading. The authors argue that the Iceland plume influence collapsed rapidly at 38 Ma and was then progressively re-established. This is an impressive study, and the authors provide high quality geochemical data and geochemical modelling to support their ideas.

Response: We thank the reviewer for this insightful and helpful review that has been used to improve the manuscript. We are pleased that the reviewer found it to be an *impressive study* and that they thought that the geochemical data and modelling is of *high quality*. Below, we provide a detailed response to each of the points raised.

Comment 1

The authors argue that changes in the mantle source composition are needed to explain the different trace element patterns in FC samples compared to VST/VSR samples. They show in Figure A3 how the trace element pattern changes between different mantle sources (i.e., DM from Workman and Hart and calibrated plume DM) if melting conditions are kept the same. But I am not convinced that the trace element differences cannot be generated by different degrees of melting. Further discussion and visualization of how the trace element pattern changes if melting conditions are changed for a constant mantle source could clarify this.

Response:

Thank you for this useful comment. Variable degrees of melting of spinel lherzolite (depleted MORB mantle) do not change the slope of REEs (i.e., La/Sm or Ce/Yb) until the mantle potential temperature is high enough such that a significant fraction of melting occurs in the garnet stability field. Basalt compositions in this study mostly reflect melting above the spinel-garnet transition at ~69–78 km and so variable melt fraction is expected to result in approximately uniform REE enrichment/depletion. The relevant Appendix figure (now Figure A2), has been updated to show the modelled change in REE concentration for changing melt degree of a constant mantle source. Regardless of melt degree, it is not possible to obtain a data fit for the VSR/VST sites using the average depleted MORB mantle (DMM) of Workman & Hart (2005) and it is also not possible to obtain a good fit for Site FC using the calibrated plume depleted mantle source, which is comparatively depleted in light REEs. This finding suggests that a change in mantle source composition is required when considering Site FC versus the V-shaped ridge/trough samples, which reflects changing plume influence.

Comment 2

Both FC samples and SGFZ samples are erupted close to transform faults. In lines 288-289 the authors note “In addition, dredge samples immediately south of the GFZ were removed to avoid atypical melting conditions affected by the transform fault.” Could the differences in trace element geochemistry between FC samples and VST/VSR samples be related to different crustal environments? The only samples that have similar trace element geochemistry to FC samples are the SGFZ samples that also erupted close to a transform fault.

Response:

We do not think that the differences between the two sites is related to higher level crustal architecture. Samples very close to the southern transform fault of the CGFZ (Charlie-Gibbs fracture zone) were removed from the database prior to PCA because transform faults can exert a minor cooling effect on the adjacent lithospheric mantle thicknesses, leading to possible anomalous enrichment [1]. Dredge samples that the PCA identified as the most similar to samples from the FC site are at least 60 km from the trace of the southern transform fault within the CGFZ. 60 km is typically thought to be enough clearance from a transform fault such that the geochemistry of dredged samples is not unduly influenced by cooling [1]. Therefore, the low melt degrees inferred from REE modelling and, critically, local crustal thickness measurements at 51.6°N are interpreted to represent ambient mantle unaffected by either a mantle plume or a transform fault.

The FC site was drilled on a segment high between fracture zones (Fig. 1a). These fracture zones are thought to be the traces of small transform faults along the ridge axis, which were only sustained during the low temperature interval following plume collapse and prior to resurgence. When the plume outflow underwent resurgence, the increase in mantle temperature and thickening of the oceanic crust inhibited the ability of the crust/spreading system to support transform faults and offsets were reduced to reform a linear axis [2–4]. These transform faults are likely to have only offset the ridge axis by a few km, given their close spacing, and therefore were probably not ‘transform faults’ in the strict sense (which would require > 30 km of offset [5, 6]). Certainly, they were not on the same order of magnitude as large tectonic-scale transform faults such as the CGFZ which has an offset in excess of 300 km, and are likely to have had a minimal impact on the thermal mantle regime.

The existence and stability of small transforms/fracture zones at the Reykjanes Ridge depends on the mantle temperature and mantle plume influence in the North Atlantic mantle [2–4]. Coupling between plume influence and the stability of the ‘transform faults’ is required by the fact that they never existed north of ~62°N, they were eliminated in a time transgressive manner away from Iceland coincident by the growth of VSR anomalies, and their elimination process correlates with basalt radiogenic isotopic enrichment implying a broad dynamic control (Fig. 3; Fig. 5).

In the revised manuscript, samples south of the CGFZ at 51.6°N are no longer referred to as SGFZ samples, which we think is misleading. Instead, we more correctly refer to the mid-Atlantic Ridge at this latitude. Text has been added to explain that we believe these samples are far enough from the CGFZ such that their chemistry is not affected by anomalous cooling effects. In the end, our strongest suit is the excellent agreement between geochemical modelling and independent crustal thickness measurements at these different sites.

Comment 3

Some of the data references for previously published data are not clear. For example, data references for both trace elements and isotopic compositions for SGFZ samples are unclear. Also, the authors state that they use DMM composition from Workman and Hart (2005), but it is not clear which DMM composition (i.e., average DMM, enriched DMM or depleted DMM).

Response:

We readily agree that it is important to provide further details on the literature which we had previously omitted. We have now amended both the main text and Methods section as well as the Source Data file to include the literature data sources. We exploited average DMM from Workman & Hart (2005), which has been added to the Methods section.

For trace element data used during PCA:

REE measurements from all sites were combined with previously published datasets from axial dredges. The compiled studies consist of Gale et al. (2013) [7], Murton et al. (2002) [8], and Jones et al. (2014) [9]. Duplicate analyses were removed. Dredge samples less than 60 km south of the southern transform fault of the CGFZ were removed to mitigate the effects of thermal cooling caused by presence of the transform fault [1]. Data sources are provided in Source Data.

For radiogenic isotope data, the bulk of measurements were compiled from Blichert-Toft et al. (2005) [10], Jones et al. (2014) [9], and Thirlwall et al. (2004) [11]. The only exception is the radiogenic Nd isotopic composition of samples at 51.6°N where the value is taken from the compilation of Gale et al. (2013) [7]. This exception has been clarified in the revised manuscript (and the appropriate reference added when referring to radiogenic isotope composition of samples at 51.6°N):

To gauge whether or not this pattern of depletion has remained stable as a function of time, radiogenic Nd isotopic compositions of whole-rock samples from each of the IODP sites have been measured and compared with an axial dredge dataset compiled by Blichert-Toft et al. (2005) [10], Jones et al. (2014) [9], and Thirlwall et al. (2004) [11].

Comment 4

Figure 2 Panel a – The authors state in lines 43-44 that samples south of 61.2°N are depleted relative to N-MORB and D-MORB. Adding N-MORB and D-MORB compositions to Figure 2a would help the reader visualize this. VST-2 is labelled twice in panel 2a both in light blue and dark blue. Light blue label is probably incorrect. Panel c – Plotting the lowest distance on top of higher distances might help this figure. Some of the samples along the ridge show relatively low distances but they are overlain by samples with higher distances

Response:

Thank you for these detailed comments on Figure 2. We have revised this figure to improve clarity and address your comments. We have also added a marker for the average composition of D-MORB to panel a and fixed the colour mix-up for VST-2. On panel c (now panel d), we agree that the on-axis samples were unclear and overlapping. To address this lack of clarity, we have now binned the ridge-axis samples every 0.2° and displayed the average distance in PC space from FC. In addition, we have added the average Ce/Yb composition of the on-axis VST-1 composition to panel a as well as its PCA position in panel b to allow comparison with the drilled VST-1 composition and aid the interpretation of Figure 3.

Comment 5

Figure 3 The most enriched seamounts on the ridge-axis are not shown in Figure 3. The $^{143}\text{Nd}/^{144}\text{Nd}$ data from the enriched VST-2 seamount is still shown in this plot but this locality is excluded from the trace element modelling because of enrichment in trace element composition. This biases the $^{143}\text{Nd}/^{144}\text{Nd}$ of the ridge-axis dredges towards more homogenous depleted $^{143}\text{Nd}/^{144}\text{Nd}$ and ignores small-scale heterogeneity in the mantle source along the Mid-Atlantic ridge. While the full variability of the VST/VSR $^{143}\text{Nd}/^{144}\text{Nd}$ data is shown.

Response:

We have now added the radiogenic isotope composition of dredges from enriched seamount 14D to this figure. This addition is an important example of small-scale heterogeneity in the mantle source along the mid-Atlantic ridge. However, the $^{143}\text{Nd}/^{144}\text{Nd}$ composition of the Reykjanes Ridge has been shown to be largely homogenous [8–10]. The flowline variation in radiogenic isotope composition from site-to-site presented by this study is large and both VST-2 (i.e. a suspected seamount) and VSR-3 have a lower $^{143}\text{Nd}/^{144}\text{Nd}$ than even the most enriched seamount from along the Reykjanes Ridge.

Comment 6

Figure 4 The classification of VST-1 samples into on axis VST-1 and VST-1 confused me because it has not been used previously in the paper. This made me wonder if on axis VST-1 and VST-1 plot in the same space on the PCA plot, and if they have similar $^{143}\text{Nd}/^{144}\text{Nd}$ composition. It would be good to see this classification also in Figures 2 and 3. Also, on axis VST-1 have a very similar color to VST-2 in Figures 2 and 3, making it is easy to confuse the two groups together.

Response:

We agree with these detailed comments about Figure 4. The first mention of ‘on-axis VST-1’ should be positioned earlier in the manuscript. Therefore, we have added the average composition to Figure 2a and b. The composition of the on-axis VST-1 is similar to the composition of drilled VST-1, and their positions overlap in REE PCA space. We have changed the colour of the axial VST-1 to a dark green to clearly distinguish it from navy blue VST-2. We have also promoted Figure A1 to the main text as panel c in Figure 2, which will help readers understand the context of the PCA results in a petrological framework.

Authors' Response to Reviewer 2

General Comments. This paper explores an interesting question regarding the temporal continuity of the Iceland Mantle Plume using an important new dataset obtained via IODP cruise 395. Given the paper's ability to link this geochemical data to some of the down-core geophysical observations, it is a useful study with a potentially impactful discussion and therefore should be published. However, I believe there are a number of items that need to be addressed prior to publication so I recommend revisions.

Response: We are grateful to Emma Watts for providing this insightful and helpful review. She states that this study is *useful* and *should be published*. We have addressed all of the items that have been raised below, which we believe have improved the manuscript significantly.

Comment 1

When looking at the dataset provided as supplementary material I have noticed a few issues with the analyses. I am slightly concerned that some of the standard reference material values are outside what is deemed acceptable. The authors should look into this in more detail and investigate why this might be.

Response:

We agree. This important point has led us to improve the manuscript and data presentation. We have now expanded the Source Data file to include the accuracy and precision of all standard measurements, allowing readers to assess the quality and reproducibility of measurements. We have also significantly expanded the Methods section to include details of our analyses and geochemical processing procedures. With regard to reproducibility, we consider each data category below. Essential details have also been added to the methods. In response to the reviewer's particular concern with regards to EPMA measurements, we now supply data that has been corrected to the reference composition of the VG-2 glass standard, which improves accuracy on major element oxides to within 2 s.d. of the alternative secondary standard.

- We note that **XRF** data is not directly exploited in this study but is provided for reference. The accuracy and precision of **XRF analysis** was determined by repeat measurements of USGS international standard materials BIR-1 and BE-N. For both standards, the accuracy of major elements is better than 1%, with the exception of Na₂O which has an accuracy $\leq 5\%$. Precision of all major elements on BE-N is better than 1%, although precision is poorer for elements near the detection limit. The precision on BIR-1 is typically $\leq 1\%$, but is slightly higher for Na₂O,

P_2O_5 , and K_2O . These data indicate good reproducibility, and are consistent with previously published XRF analyses carried out on equivalent basaltic materials [12, 13]. All measurements are available in the corrected and updated Source Data file.

- Accuracy and precision of **ICP-MS** was monitored by repeat measurements of external USGS reference materials BCR-2, BHVO-2, and BIR-1 throughout the analytical session. Across all standards, the accuracy of REEs was better than 3%, and typically better than 2%. Precision was also typically better than 3%. Full accuracy and precision statistics of standard analyses are provided in the Source Data file. These results are as expected for solution ICP-MS analysis and imply good external reproducibility.
- To ensure reproducibility of the **radiogenic isotope ratio analyses**, United States Geological Survey (USGS) rock reference materials BCR-2 and BHVO-2 were taken from powders to Nd fraction and analysed simultaneously with the samples. These standards were measured with $^{143}\text{Nd}/^{144}\text{Nd}$ of 0.512641 ± 13 (2 s.d., $n=4$) for BCR-2 and $^{143}\text{Nd}/^{144}\text{Nd}$ of 0.512993 ± 15 (2 s.d., $n=4$) for BHVO-2. These measurements compare to the GeoReM reference values of 0.512635 ± 29 (1 s.d.) and 0.512979 ± 14 (1 s.d.) for BCR-2 and BHVO-2, respectively, implying good external reproducibility. All procedural blanks contained negligible Nd. Similar reproducibility was obtained for radiogenic isotope analyses for samples run in 2024 (see revised Methods).
- Accuracy and precision of **EPMA** measurements were monitored by regular measurement of external standard glasses VG-2 and NMNH 113716-1 from the Smithsonian Museum of Natural History collection. Accuracy for all major element oxides with respect to VG-2 was $\leq 5\%$ and was better than 2% for SiO_2 , MgO and FeO which are the only major element oxides used in this study. Accuracy on NMNH 113716-1 was $\leq 5\%$ for all major element oxides except Na_2O and P_2O_5 ; accuracy was $\leq 3.5\%$ for SiO_2 , MgO and FeO. The precision on both standards was $\leq 3\%$ (1 s.d.) for all major element oxides of abundance not near detection limit. Repeat standard measurements were monitored for analytical drift and significant variation was not observed. This performance is overall comparable with other published studies that analysed these standards [14–17], and other studies employing EPMA on natural glasses [12, 13], and was deemed adequate for the purposes of our study. Uncertainty is not provided on the reference standard values [18], however variation in reported values from a number of published studies is $\leq 3\%$ for VG-2 and up to 7% for NMNH 113716-1 [14–16, 18]. In particular, studies repeatedly report Na_2O values for NMNH 113716-1 greater than 5% higher than the preferred value. Analysis of standards and compiled results are provided in the Source Data file.
- To maximise the reproducibility and to facilitate future use of generated data, major element data that have been corrected against the preferred composition of VG-2 [18] is also provided in the Source data file, along with the factors applied for correction. This correction improves

the accuracy of major element oxides on the NMNH 113716-1 standard, to $\leq 2\%$ for all oxides above 0.5 wt.%, with the exception of Na_2O which as discussed above does not have a reliable preferred concentration reported. This correction procedure follows the method of Helz (2021) [19], and is similar to the inter-laboratory bias corrections applied by Gale et al. (2013) [7] on the VG-2 glass standard.

- Accuracy and precision of **LA-ICP-MS** analysis was monitored by repeat measurements of external reference materials BIR-1G and BCR-2G. Accuracy on BIR-2G is $\leq 5\%$ for most trace elements and all REEs, which are the focus of this study. Accuracy with respect to BCR-2G for REEs is $\leq 5\%$ except for Ce which is 6%. Most REEs on both standards have an accuracy $\leq 3\%$. Precision of trace element analyses was variable but better than 10% for most REEs and trace elements of sufficient concentration above the detection limit. Details of accuracy and precision of these standards are provided in Source Data file.

Comment 2

With the melt modelling (section 2.3 Geochemical modelling of Fractured Crust), you have clearly stated your input parameters in the main text (and expanded on them in the methods) which is great. However, there is no discussion of uncertainties in those and how your results would change if you varied them slightly. Please add a section (into the main text and/or methods) on the sensitivity of the model and the resulting uncertainty if the input parameters are varied slightly.

Response:

We agree with this point. It is of course important to discuss the uncertainties involved in modelling and parameter choice and to assess to what extent they affect our conclusions. We have now added an explicit piece to the Methods section, which is also shown below.

- We have carried out a temperature sensitivity analysis, varying potential temperature around the preferred value, to determine the range of temperatures that could fit these geochemical observations (see Figure 4). The range of appropriate temperatures for Site FC clearly do not overlap those of the VSR/VST sites.
- Modelling is naturally dependent upon the estimated melt productivity of the mantle for a given mantle potential temperature. Melt productivity is dependent on the choice of mantle source parameterisation and on the mantle water content. Mantle water content was found to have a very minor influence upon melt generation and melt composition—trebling mantle water for melting hydrous lherzolite from 0.01 wt% to 0.03 wt% increases the predicted crustal thickness

by just 0.2 km and forward modelled light REE concentrations change only within measurement error.

- The conclusions of this study partly rely upon the relative difference in modelled melt degree between drilled sites. It is therefore important to use a constant set of assumptions regarding the mantle source parametrisation and mantle mineralogy. Uniform changes to these assumptions will not change the relative differences and, thus, our conclusions. As a test, we changed the lherzolite melting parametrisation to that of Ball et al. (2022) [20]. This lherzolite is less fusible than that of Katz et al. (2003) and so at the same mantle potential temperature less melt is generated, which leads to a more REE-enriched aggregated melt composition. At Site FC, modelling dictated a mantle potential temperature of $1322 \pm 15^\circ \text{C}$, which remains approximately ambient. Equivalent (within uncertainty) melt thickness is estimated, producing 6.0 km of crust (compared to 5.8 km for Katz et al. (2003)). This test suggests that conclusions drawn from our modelling are robust with respect to mantle source parameterisation.
- The effect of pyroxenite on melt production and melt composition also depends upon modelling assumptions, although mantle proportions of this component are deemed to be minor. Changing the $^{143}\text{Nd}/^{144}\text{Nd}$ of the pyroxenite component from 0.5127 [21] to 0.5129 [22] when melting at conditions modelled for Site FC, increases the predicted ε_{Nd} of the homogenised melt by 0.13, which is smaller than the typical ε_{Nd} measurement error. For modelling at VSR-3 (U1562B), which has a greater expected proportion of mantle pyroxenite (2.6%), this change increases the ε_{Nd} of the homogenised melt by 0.63. Mantle pyroxenite would be required to increase to 4.5% to match the observed ε_{Nd} . The REE melt enrichment that the additional pyroxenite results in can be offset by a $\sim 15^\circ \text{C}$ increase in mantle potential temperature. This small increase does not alter the conclusions of our study. In fact, the relative difference in temperature predicted between the FC and VSR/VST sites is actually greater.
- An alternate approach to parameterisation of the garnet-spinel transition is to assume that the transition depths are isobaric, using the pressure range constrained by the thermodynamic experiments and modelling of Tomlinson & Holland (2021) [23]: 2.23–2.52 GPa (69–78 km). Tests that use these values led to a very minor change in predicted heavy REE concentrations, requiring no change to the calculated mantle potential temperature at Site FC. For higher temperatures at, e.g. VSR-3 (U1562B), a slight increase in the proportion of melting within the garnet stability field leads to a modelled depletion of heavy REEs whilst the light REEs remain largely unchanged. To improve the fit of the modelled REEs, one option is to decrease the mantle potential temperature and slightly decrease the mass fraction of mantle pyroxenite. The mantle potential temperature determined in this way is $\sim 1400^\circ\text{C}$. This temperature remains significantly above ambient mantle and that predicted for Site FC, and therefore the overall conclusions of this study are unaffected.

Comment 3

I believe that a few more markers for the readers towards figures, methods section, appendixes, supplementary data, will make it a lot clearer and easier for readers to follow, and may help them realise there is a dataset they could utilize (leading to more citations).

Response:

We readily agree and we have added more markers towards figures, methods, appendix and Source Data file to improve readability.

Comment 4

In the abstract add in the age after ‘continental breakup’ in sentence #4. Now, it isn’t clear the timelines that this has occurred over.

Response:

We agree and we have added an appropriate age citing the chronological compilation of Wilkinson et al. (2016) [24].

Comment 5

Please consider splitting sentence #5 up as its meaning and impact gets a little lost given the length and complexity of the sentence.

Response:

We agree and we have split up the sentence as follows:

Recovered ~ 32 Ma basalt samples have rare earth element compositions equivalent to mid-Atlantic Ridge dredge samples located south of the present-day plume influence. These compositions can be modelled by passive upwelling and melting of depleted MORB mantle with a potential temperature of $\sim 1300^\circ$ C.

Comment 6

Line 12: change 'may' to 'are suspected' as the use of may sounds like you will put an alternative hypothesis after but there is none stated.

Response:

We agree and we have amended this sentence.

Comment 7

Line 13: The sentence with the list of boreholes and ages is not clear. Please clarify what the ages are of? Are they the deepest depth of rock found in the borehole, the age of the VSR or something different? This should be made very clear to all readers.

Response:

We readily agree that it is important for the readers to have clarity on the age constraints of these sites. We have amended the main text to read:

The four boreholes that penetrate V-shaped ridges and troughs are: U1555I (i.e., VST-1) with a magnetic basement age of 2.8 Ma; U1563B (VSR-2) at 5.2 Ma; U1554F (VST-2) at 12.7 Ma; and U1562B (VSR-3) at 13.9 Ma (Figure 1a and b).

We have then expand on this issue within the Methods section when describing the age corrections applied to radiogenic isotope data:

For that sample nearest to the sediment/basement interface, age is constrained by magnetic anomaly which is then verified using palaeotological biomarkers at the base of the sediment pile. The ages are 2.8 Ma for U1555I, 5.2 Ma for U1562B, 12.7 Ma for U1554F, 13.9 Ma for U1562B, and 32.4 Ma for U1564F. In this study, it is assumed that there is no significant age variation within the basalt drilled section.

Comment 8

Line 17: Is FC the name of the borehole? Try to clarify this please

Response:

It is the abbreviation that we use for this site. We have amended the sentence to improve clarification as follows:

A fifth borehole (U1564F hereinafter referred to as FC) targeted much older (i.e., 32.4 Ma) rugose crust that is devoid of V-shaped ridges and troughs but instead characterized by extensive fracture zones (Figure 1a and b).

Comment 9

Line 24: Add in rough age of Neogene to aid readers.

Response:

We have amended this sentence to be more specific about the time frame of hypothesised resurgence:

Reversion to smooth (i.e., not fractured) crustal generation with development of V-shaped ridges and troughs may imply plume resurgence with concomitant increase in volume flux from ~ 30 Ma to the present day [25]...

Comment 10

Line 27: Please state that you're using radiogenic isotopes and expand or give examples on what you mean by contextual geophysical observations.

Response:

We agree and we have amended the sentence as follows:

Here, we wish to test these alternative hypotheses by exploiting trace element and radiogenic isotopic measurements from all five sites. These measurements are combined with crustal thickness constraints from modern seismic wide-angle and legacy refraction surveys in order to reconstruct the influence of the Iceland plume during Cenozoic times.

Comment 11

Line 32: Please include a date of continental break-up for context.

Response:

We agree and we have added an appropriate age citing the dating compilation of Wilkinson et al. (2016) [24]

Comment 12

Line 35: Could you be more specific about when the plumes influence re-established?

Response:

We agree that we should be more specific here. We have now amended the sentence as follows and included an appropriate reference:

At ~ 30 Ma, plume influence started to re-establish itself, which may reflect migration of the mid-ocean ridge towards the plume conduit (Poore et al., 2009).

Comment 13

Line 40: Why was whole rock ICP-MS only done on the FC sample? This is not explained or justified in the text. If you can't fit an explanation in the main paper for word limit reasons, please include in the methods section and point the readers to that section.

Response:

We agree that this choice should be justified in the text. We have now added an explanation to the methods section and included a marker to the methods in the main text. We have added to the methods:

Here, we present whole-rock major and trace element data for hole U1564F at Site FC only, which supplements limited glass availability.

We note to the reviewer that ICP-MS analysis was also conducted on whole-rock material from the VSR/VST sites. These measurements, particularly for REEs which is the focus of this study, are closely similar to glass data and yield the same modelling outcomes. However, glass data is typically preferred especially for elements that are fluid mobile and sensitive to sea-water/hydrothermal alteration. This whole-rock ICP-MS data is not presented here because it is being applied to a separate publication that is answering a distinct question to the one addressed in this manuscript (White et al., *sub judice*).

Comment 14

Line 41: It's great that you have used previously published REE data from dredged samples to be able to make further conclusions. However, there is no discussion of that data in the methods section. Please amend this and add in a section on that data, and the limitations of it given that it is dredged data and will therefore have more of a spatial uncertainty etc. Given this study looks into the spatial and temporal changes of the Iceland plume, the (age and location) uncertainties in all data should be acknowledged and discussed.

Response:

We readily agree that it is important to provide further detail on legacy measurements exploited here. We have now amended both the main text and Methods section, as well as provided all compiled data within the Source Data file. We agree that there is a greater spatial uncertainty associated with axial dredge material, but we are reassured by the dense sampling and comprehensive existing characterisation of the along-axis geochemical trends. The Methods section now reads:

REE measurements from all sites were combined with previously published datasets from axial dredges. The compiled studies consist of Gale et al. (2013) [7], Murton et al. (2002) [8], and Jones et al. (2014) [9]. Duplicate analyses were removed. Dredge samples less than 60 km south of the southern transform fault of the CGFZ were removed to mitigate the effects of thermal cooling caused by presence of the transform fault [1]. Data sources are provided in Source Data. Dredge materials have inherently greater spatial uncertainty compared to ocean drilling. This uncertainty is partly addressed by the high axial sample density combined with the fact that axial geochemical trends in trace element and radiogenic isotope composition are well characterised [7–10].

Comment 15

Line 45: Please point the reader to Figure 2 when discussing the GFZ.

Response:

We agree. Given emphasis is placed on the similarity in REE composition between FC and samples south of the Charlie-Gibbs FZ, we have now altered Figure 1 to extend further south. We have also added a marker to Figures 1 and 2 at the relevant part of the text.

Comment 16

Line 61: Lower than what? Please be clear in what you are comparing to

Response:

We agree and we have now amended this sentence to read:

Co-location within P_{1r} - P_{2r} space of Site FC samples and mid-Atlantic Ridge samples at 51.6° N reflects uniform REE enrichment compared with samples which lie within the Icelandic enrichment gradient at $\sim 62^\circ$ N. This enrichment, represented by an increase in P_{1r} score, can be modelled by a decrease in mantle potential temperature of $50 \pm 25^\circ$ C or, alternatively, by 19% of olivine crystallisation (Figure 2c).

We hope that this concept is clarified by the promotion of Figure A1 into the main text as Figure 2c.

Comment 17

Line 78: Add in the words previously and radiogenic to make the sentence read: ‘Along the Reykjanes Ridge, the observed gradient in radiogenic isotopic enrichment has previously been interpreted as progressive dilation of an enriched plume source by mixing with depleted mantle during southward outflow from Iceland [23] (Fig. 3a).’

Response:

We agree and we have accordingly amended this sentence.

Comment 18

Lines 87-92: This is very interesting to see the variation with time. I think it would be even more helpful to the community if you could also assess the ^{87}Sr - ^{86}Sr or $^{206}\text{Pb}/^{204}\text{Pb}$ with time. This would also then allow to look in isotopic-isotopic space e.g. ^{87}Sr - ^{86}Sr vs $^{143}\text{Nd}/^{144}\text{Nd}$ and get a better grasp of the mantle end-members contributing and allow for easier discussion of the pyroxenites etc. I understand that would require a lot of work, but I think it could significantly help the study go to the next level of discussion.

Response:

We agree that this additional would be extremely helpful. However, measuring these radiogenic isotope systems is beyond the scope and resources of this particular study. Along the Reykjanes Ridge axis,

excellent correlations exist between Sr-Nd-Pb-Hf radiogenic isotope compositions [10], and so it is likely that the significant change in radiogenic Nd presented here would also correlate with further radiogenic isotope systems. Therefore, for the purposes of our present study, we believe that radiogenic Nd sufficiently captures the likely mantle source variation across the North Atlantic region with time. Clearly, this proposed analysis is a promising avenue for further study.

Comment 19

Lines 106-108: When stating you think that the REE compositions and crustal thickness provide a more robust measure of plume influence - what statistics are used to back this up with? If this is a sweeping statement without any method of it being a 'measure' please amend the sentence. If it can be used as a mathematical predictor this needs to be expanded on and explained, including the limitations.

Response:

We agree and we acknowledge that we should have been careful describing this agreement as a 'measure'. We believe mantle potential temperature together with depth and degree of melting, are better indicators of the presence of plume influence compared to traditional mantle source proxies such as ΔNb and ε_{Nd} . This view prevails in the North Atlantic Ocean where traditionally interpreted 'non-plume' source signatures ($\Delta\text{Nb} < 0$ and $\varepsilon_{\text{Nd}} \sim 10$) occur where there is clearly plume influence as manifest by thick oceanic crust and by anomalously high degrees of mantle melting. We have therefore amended the text as follows:

Thus, source proxies may not necessarily be reliable indicators of the presence or absence of plume influence. Instead, we suggest that the depth and degree of isentropic melting at the mid-oceanic ridge, which can be inferred by combined modelling of trace element concentrations and oceanic crustal thickness, is a clearer indicator of plume influence.

Comment 20

Line 117: How much is trace pyroxenite? For reproducibility, please state a numerical value.

Response:

We agree and we have now amended this sentence to include the percentage of mantle pyroxenite.

Comment 21

Line 134: If you believe that VST-2 drilled a seamount, why was it included in the first place?

Response:

Hole U1554F was chosen and drilled by IODP with a view to targeting VST-2. However, geochemical analysis revealed that its composition was highly anomalous, and similar to isolated trace element enriched seamount dredges from the Reykjanes Ridge axis (e.g. the 14D ‘mega-seamount’ [8, 26]), and to rocks sampled from the onshore Reykjanes Peninsula. Subsequent re-examination of orthogonal seismic reflection profiles at the site suggests that the drilling may have penetrated a small seamount. Considering that this drilling site represented an important part of the scientific agenda for IODP Expedition 395 we believe it would be confusing to those aware of the Expedition objectives to omit it entirely from this manuscript. Therefore, we present its geochemical data for completeness but we do not attempt to model it here due to the inherent complexity of modelling heterogeneous enriched mantle melts (it is considered in more detail by White et al., *sub judice*).

With respect to the uncertainty concerning seamount characterisation, we have amended the sentence as follows:

Therefore, borehole U1554F may have penetrated an isolated seamount or been influenced by localised mantle heterogeneity, which means that its composition might not be representative of this particular V-shaped trough [27]. It is not considered any further in this contribution.

Comment 22

Line 136: While I agree some of the variability aligns along the Dy axis on the PCA, however it appears the trend doesn't fully line up and so could be influenced by Gd and Lu too (especially the cream Reykjanes Ridge samples). It might be worth looking at the best fitting linear correlation of the trend (s) for the whole dataset and the individual groups to see if they agree with your statement and if they're statistically significant. This will make your statements more robust.

Response:

It is evident that the variability does not fully align with the Dy vector and so we have accordingly amended the sentence. The depth of melting (and therefore volume and stability of garnet in the mantle melting domain) will affect all heavy REEs (Dy, Yb, Lu especially). The degree of melting and extent of fractional crystallisation tend to align along the Gd vector in this PCA (until the degree of

melting is high enough that it begins to invoke garnet field melting; Figure 2c). All of these processes vary from samples to sample and so the trend of groups of samples reflects a combination of processes, weighted by their relative importance. The sentence now reads as follows:

This array is primarily aligned with the Gd-Dy-Lu vectors, which suggests that translation along the array reflects a combination of changes in depth of melting that affects heavy REEs and degree of melting and/or fractional crystallisation (Figure 2c).

Figure Comments

Comment 23

Figure 1 -Is the scale for gravity anomaly the same for both (a) and (b). If so, consider centering across the whole figure so it is below both (a) and (b). This would make it clearer to readers that the scale is the same. If it's not the same, please include a new scale for (b). - In subfigure (a) The purple lines are quite hard to see on the red and blue background. Please see if there is a way to make them clearer.

Response:

We agree and we have now considerably revised this figure to improve its clarity and impact. We have added an additional panel to allow incremental addition of detail. In this way, the gravity anomalies are now more clearly observed. We have also extended the latitudinal range of all panels to allow incorporation of the region south of the Charlie Gibbs Fracture Zone. Purple lines for seismic experiments have been changed to a bright green, which we believe is easier to see. Crustal thickness values have also been added to panel (c) to aid reader interpretation.

Comment 24

Figure 2: With the Charlie-Gibbs Fracture Zone – please be consistent with the acronym used to refer to it by. In the main text (Line 45) you state it is GFZ, then in the legend you use CGFZ. Please include error bars on your plots of if the error is smaller than the data points state that in the caption. When using different colours for datapoints please also consider using different shapes as it may help if people are colourblind or print in black and white. In Fig. 2c please make it clearer exactly where the GFZ and BFZ are. At the moment, it is a little ambiguous.

Response:

We agree. We have now amended all acronyms for the Charlie-Gibbs Fracture Zone to CGFZ. PCA itself has no inherent error as it is a dimensionality reduction technique applied to a given dataset. The REE measurements exploited for PCA do have uncertainty. However, this information is not provided within the Gale et al. (2013), Jones et al. (2014) or Murton et al. (2002) database and so cannot be propagated into panel b or provided on panel a. The uncertainty in the composition of the VSR/VST/FC sites can be estimated from the range of samples analysed from each drilling location. We have also updated the figure to use different shapes as well as colours where appropriate. Finally, we have added dotted traces to the bathymetry map on panel d, indicating the positions of the BFZ and CGFZ.

Additional changes have been made to this figure including promoting Figure A1 to panel c since it is illuminating to place PCA results within a petrological context. In addition, on panel d we have now binned the axial samples every 0.2° of latitude and display the average distance within PC space from the FC site. We have added the average Ce/Yb ratio of the axial VST-1 composition to panels a and b, which permits comparison with the drilled VST-1 composition thus aiding understanding of Figure 3.

Comment 25

Figure 4: Please label the grey box with crustal thickness (I'm guessing) to signify its meaning.

Response:

We have added this description to the figure caption but we omit it from the figure itself to limit visual clutter and distraction.

Comment 26

Figure 5: Please consider adding what the yellow dot represents to the figure itself as well as the caption. The same applies to the green arrows.

Response:

This figure has been considerably revised to improve clarity. We have now added annotations for outflow vectors as well as yellow dots. We hope that the changes made to this figure make our intent clearer.

Methods Comments

Comment 27

Section 5.2.1: When discussing the Nd isotopes please write the isotope ratio out in full as not all readers will know the pair.

Response:

We agree and we have amended all instances within the text.

Comment 28

Also, paragraph 1 in this section (5.2.1) jumps around a bit making it hard for the reader to follow. Please reorder and keep procedures more logical and in order of actual steps. Consider discussing trace element analysis first, then move on to $^{143}\text{Nd}/^{144}\text{Nd}$ analysis.

Response:

We agree. We have now reorganised this part of the Methods section, splitting trace element analysis and radiogenic isotope analysis into separate sections.

Comment 29

With the provided data spreadsheet. Please include your standard deviation of your measurements for the standards e.g. BIR-1 as well as the range or standard deviation of the accepted values so that they can be compared. Currently your SiO₂, MgO and CaO values of BIR-1 appear to be slightly outside of the accepted value stated.

Response:

We agree and we have now updated the Source Data spreadsheet to include standard deviations of repeat measurements of standards as well as the uncertainty of the reference value where available. We have also added estimates of precision (%RSD) and accuracy to all data types.

BIR-1 is a reference material that was analysed during XRF work (i.e. in addition to BE-N). We note that XRF data are not actually exploited in this manuscript and are provided for reference only. For both standards, the accuracy of major elements is better than $\sim 1\%$, with the exception of Na₂O which has an accuracy $\leq 5\%$. Although some elements still strictly have average values that are marginally outside the stated reference value, these measurements are coherent with previously published XRF analyses on similar basaltic materials [12, 13]. Interested users that require higher accuracy on Na₂O can always apply a correction based upon the reported reference compositions.

Comment 30

Line 218: What type of beads were made for XRF? Was it lithium tetraborate or something different? What ratio was used of that to your sample? More detail is required for the results to be reproducible.

Response:

We agree with the need for clarity on this and related issues. We have now extensively expanded the method and processing description for all analyses to ensure clarity and to facilitate reproducibility.

Comment 31

Line 233: Please add in the accepted values for the rock reference materials BCR-2 and BHVO-2 and the appropriate reference.

Response:

We agree and we have now added accepted reference values from the GeoReM database.

Comment 32

Line 238: While I understand you are following the sample writing pattern for other analyses but stating a standard deviation when your $n = 2$ is statistically meaningless. Please amend and just write both values.

Response:

We agree and we have amended this to state the average with a quoted error of half the measured range:

BCR-2 and BHVO-2 USGS standards were measured with $^{143}\text{Nd}/^{144}\text{Nd}$ of 0.512633 ± 7 ($n=3$) for BCR-2 and $^{143}\text{Nd}/^{144}\text{Nd}$ of 0.512985 ± 4 ($n=2$) for BHVO-2

Comment 33

Line 248: Please consider putting 6M in brackets next to your HCl concentration to aid readers.

Response:

We agree and we have added 6M in parentheses to aid readers, as opposed to using 6N which is equivalent but may be unknown to some readers.

Comment 34

Line 270: Please add a reference for the Glitter software used.

Response:

We agree and we have accordingly amended text to provide the website link whence software information can be found together with relevant literature reference.

Comment 35

Section 5.2.2: When discussing the EMPA and LA-ICP-MS analysis it is clear you have done multiple analyses per sample which is good however more detail in the data spreadsheet is needed. I understand you have summarized the data putting the average major and trace element compositions for each sample on one line however those do not show the number of spots for each nor the standard deviation. This should be the minimum detail included. Ideally you would have this and, on another sheet, you would have the raw data provided so that it may be used by others and is completely transparent.

Response:

We readily agree and we have acted to increase the level of supporting detail. We have included the number of chips and spots involved in the calculation of each sample average to the Source data spreadsheet. We also supply an additional piece about the average standard deviation of spot measurements across chips as well as the standard deviation of chip averages used to calculate the sample mean. A detailed description of the filtering and averaging procedures applied to both EPMA and LA-ICP-MS analysis has been added to the Methods section. In the Source Data, we now also provide the raw data for individual points to allow users to apply their own bespoke filtering and averaging method.

Comment 36

Also, it is not clear when the standards for any of the analysis were carried out. Ideally, they would have been spaced throughout the run with the minimum being at the start and end of the run to check for analytical drift. Please provide more information on this.

Response:

We readily agree and we have now substantially increased the level of detail provided for each analysis type. In all cases, external reference materials were analysed before, during and after analytical sessions. They were also monitored for analytical drift. In the Source Data file, we have added raw standard runs as well as the overall average and standard deviations.

Comment 37

Finally, your EMPA VG-2 standard values for SiO₂, Al₂O₃ and TiO₂ and NMNH 113716-1 values for SiO₂, Na₂O and TiO₂ are outside of the accepted values. Please go back and investigate this as it is not mentioned at all in either the methods or paper but it is very important as it puts doubts on your values of those elements.

Response:

We agree that we have not been sufficiently clear on this point. We have detailed a response earlier in this rebuttal document but we summarise the salient points again here. This information is also provided for readers in the revised Methods section.

- The accuracy for all major element oxides on VG-2 was $\leq 5\%$, and was better than 2% for MgO and FeO which are the only major element oxides exploited in this study. The accuracy for NMNH 113716-1 was $\leq 5\%$ for all major element oxides except Na₂O and P₂O₅; accuracy was $\leq 3.5\%$ for SiO₂, MgO and FeO.
- Whilst some of the reported averages lie slightly outside the stated reference value, overall this accuracy is comparable with other published studies that also analysed these standards [14–17]. This observation is also true of other studies employing EPMA on natural glasses [12, 13] and so it was considered suitable for the purposes of our study.
- This comparison is made more challenging since uncertainty is not provided with respect to the reference standard values [18]. However, variation in reported accuracy from several published studies is $\leq 3\%$ for VG-2 and up to 7% for NMNH 113716-1 [14–16, 18]. In particular, studies repeatedly report Na₂O values for NMNH 113716-1 that are more than 5% higher than the accepted value.

- Nevertheless, whilst we considered the accuracy of our EPMA analysis to be sufficient for our purposes, we recognise that some readers may require greater accuracy. Therefore, we also provide major element oxide data that has been corrected with respect to the accepted composition of *VG-2* [18]. This correction improves the accuracy of major element oxides for the NMNH 113716-1 standard to $\leq 2\%$ for all oxides above 0.5 wt.%, with the exception of Na_2O , which, as discussed above, does not have a reliable preferred concentration reported. This correction procedure follows the method of Helz (2021) [19]. It is similar to the inter-laboratory bias corrections applied by Gale et al. (2013) [7] with respect to the *VG-2* glass standard.

SiO_2 concentrations of the glass chips, as measured by EPMA, are used to normalise the trace element data collected by LA-ICP-MS. This correction is proportional, so a 1% error in SiO_2 will propagate to a 1% error in trace element concentrations. EPMA analysis achieved an uncorrected accuracy of $< 2\%$, and therefore the propagated uncertainty in trace elements is 2%. This is typically smaller than the LA-ICP-MS measurement uncertainty. Trace element data calculated using raw SiO_2 concentrations, as well as trace element data calculated using SiO_2 concentrations corrected on the *VG-2* glass standard, are provided in the Supplementary data sheet. **The PCA and modelling efforts presented by this study have been updated to use the corrected trace element data.** However, identical analysis using uncorrected data yields identical conclusions. For example, using uncorrected FC glass data has a best fitting mantle potential temperature 3°C higher than corrected data, which is within the temperature range applied to fit the data range and is essentially unresolvable within melt modelling methods.

Comment 38

Line 280: Please add a space between 'position.PCA'

Response:

We agree. We have amended this sentence to read:

PCA was implemented using the decomposition module and PCA algorithm in Scikit-learn (v1.5.1).

Comment 39

Line 281: Please add a table(s) in of the PCA element loadings and principle component cores to the appendix

Response:

We agree and we have added the PCA element loadings and rotated principal component scores to a table within the Source Data.

Comment 40

Line 283: You discuss the filtering with Sr and Eu but is that the only filter condition you used? If not, please state all filtering requirements. Line 288: Please clarify those samples were removed. State what the $Sr/Y > 3$ and negative P_2 mean for the readers.

Response:

We agree that the filtering process used was not written sufficiently clearly and we have amended this as follows:

Whole-rock samples from Site FC were filtered for plagioclase accumulation since large phenocrysts were observed in some thin sections [28]. Sr/Y is an appropriate proxy for determining the extent of plagioclase crystal accumulation because Sr is highly compatible in anorthitic plagioclase compared with Y [29]. Plagioclase accumulation is also expected to elevate Al_2O_3 content and may generate a Eu anomaly. PCA was carried out on whole-rock samples from Site FC using the method as described below. P_2 was found to dominantly reflect plagioclase accumulation with correlated loadings on Sr, Al_2O_3 and Eu. Whole-rock measurements were removed if $Sr/Y > 3.2$ (glass average = 2.7), which correlates with Al_2O_3 and generally gives a P_2 score elevated with respect to the glass samples.

References

1. Langmuir, C. H. & Bender, J. F. The geochemistry of oceanic basalts in the vicinity of transform faults: observations and implications. *Earth Planet. Sci. Lett.* **69**, 107–127 (1984).
2. White, R. S., Bown, J. W. & Smallwood, J. R. The temperature of the Iceland plume and origin of outward-propagating V-shaped ridges. *J. Geol. Soc.* **152**, 1039–1045 (1995).
3. Gerya, T. Origin and models of oceanic transform faults. *Tectonophysics* **522**, 34–54 (2012).

4. Jones, S. M. Test of a ridge–plume interaction model using oceanic crustal structure around Iceland. *Earth Planet. Sci. Lett.* **208**, 205–218 (2003).
5. Carbotte, S. M., Smith, D. K., Cannat, M. & Klein, E. M. in *Magmatic rifting and active volcanism* (eds Wright, T. J., Ayele, A., Ferguson, D. J., Kidane, T. & Vye-Brown, C.) 249–295 (Geol. Soc. Lond. Spec. Publ., 2015).
6. Hey, R., Martinez, F., Höskuldsson, Á. & Benediktsdóttir, Á. Multibeam investigation of the active North Atlantic plate boundary reorganization tip. *Earth Planet. Sci. Lett.* **435**, 115–123 (2016).
7. Gale, A., Dalton, C. A., Langmuir, C. H., Su, Y. & Schilling, J.-G. The mean composition of ocean ridge basalts. *Geochem. Geophys. Geosyst.* **14**, 489–518 (2013).
8. Murton, B., Taylor, N. R. & Thirlwall, M. F. Plume–ridge interaction: a geochemical perspective from the Reykjanes Ridge. *J. Pet.* **43**, 1987–2012 (2002).
9. Jones, S. M. *et al.* A joint geochemical–geophysical record of time-dependent mantle convection south of Iceland. *Earth Planet. Sci. Lett.* **386**, 86–97 (2014).
10. Blichert-Toft, J. *et al.* Geochemical segmentation of the Mid-Atlantic Ridge north of Iceland and ridge–hot spot interaction in the North Atlantic. *Geochem. Geophys. Geosyst.* **6** (2005).
11. Thirlwall, M. F., Gee, M. A. M., Taylor, R. N. & Murton, B. J. Mantle components in Iceland and adjacent ridges investigated using double-spike Pb isotope ratios. *Geochim. Cosmochim. Acta* **68**, 361–386 (2004).
12. Neave, D. A., MacLennan, J., Hartley, M. E., Edmonds, M. & Thordarson, T. Crystal storage and transfer in basaltic systems: the Skuggafjöll eruption, Iceland. *J. Petrology* **55**, 2311–2346 (2014).
13. Passmore, E., MacLennan, J., Fitton, G. & Thordarson, T. Mush disaggregation in basaltic magma chambers: evidence from the AD 1783 Laki eruption. *J. Petrology* **53**, 2593–2623 (2012).
14. Guyett, P. C. *et al.* Optimizing SEM-EDX for fast, high-quality and non-destructive elemental analysis of glass. *J. Anal. At. Spectrom.* **39**, 2565–2579 (2024).
15. Thornber, C. R. *et al.* *Whole-rock and glass major-element geochemistry of Kilauea Volcano, Hawaii, near-vent eruptive products: September 1994 through September 2001* tech. rep. (U.S. Geol. Surv., 2002).
16. Johnson, A., Dasgupta, R., Costin, G. & Tsuno, K. Electron probe microanalysis of trace sulfur in experimental basaltic glasses and silicate minerals. *Am. Mineral.* **109**, 2162–2172 (2024).
17. Ro, S. *et al.* Origin of the 1458/59 CE volcanic eruption revealed through analysis of glass shards in the firn core from Antarctic Vostok station. *Commun. Earth Environ.* **6**, 828 (2025).

18. Jarosewich, E., Nelen, J. A. & Norberg, J. A. Reference samples for electron microprobe analysis. *Geostand. Newslett.* **4**, 43–47 (1980).
19. Helz, R. T. Major-element compositional data and thermal data for drill core from Kīlauea Iki lava lake, plus analyses of glasses from scoria of the 1959 summit eruption of Kīlauea Volcano, Hawaii. *Open-File Rep. U.S. Geol. Surv.* **1012** (2020).
20. Ball, P. W., Duvernay, T. & Davies, D. R. A coupled geochemical-geodynamic approach for predicting mantle melting in space and time. *Geochem. Geophys. Geosyst.* **23**, e2022GC010421 (2022).
21. Koornneef, J. M. *et al.* Melting of a two-component source beneath Iceland. *Journal of Petrology* **53**, 127–157 (2012).
22. Rudge, J. F., Maclennan, J. & Stracke, A. The geochemical consequences of mixing melts from a heterogeneous mantle. *Geochim. Cosmochim. Acta* **114**, 112–143 (2013).
23. Tomlinson, E. L. & Holland, T. J. B. A thermodynamic model for the subsolidus evolution and melting of peridotite. *J. Pet.* **62**, egab012 (2021).
24. Wilkinson, C. M., Ganerød, M., Hendriks, B. W. H. & Eide, E. A. in *The NE Atlantic Region: A Reappraisal of Crustal Structure, Tectonostratigraphy and Magmatic Evolution* (eds Péron-Pinvidic, G. *et al.*) First published online November 8, 2016 (Geol. Soc. Lond. Spec. Publ., 2016).
25. Poore, H. R., White, N. & Jones, S. A Neogene chronology of Iceland plume activity from V-shaped ridges. *Earth Planet. Sci. Lett.* **283**, 1–13 (2009).
26. Shorttle, O. *et al.* Fe-XANES analyses of Reykjanes Ridge basalts: Implications for oceanic crust's role in the solid Earth oxygen cycle. *Earth Planet. Sci. Lett.* **427**, 272–285 (2015).
27. White, N. J. *et al.* Geochemical Significance of Diachronous V-Shaped Ridges and Troughs that flank Reykjanes Ridge South of Iceland. *in preparation* (2025).
28. Parnell-Turner, R. E. *et al.* in *Proceedings of the International Ocean Discovery Program: Reykjanes Mantle Convection and Climate* (ed JRSO, I.) 0-49 (IODP Publications, Texas, 2025).
29. Shi, H., Xia, Y., Xu, X., Zhu, J. & He, J. Crystal-melt separation of the Cretaceous volcanic-plutonic complex in SE China: High Sr/Y rocks generated by plagioclase accumulation. *Lithos* **430**, 106848 (2022).